# Physiological adaptations and myocellular stress in short-term, high-frequency blood flow restriction training: A scoping review

Victor Sabino de Queiros[1], Nicholas Rolnick[2], Phelipe Wilde de Alcântara Varela[1], Breno Guilherme de Araújo Tinôco Cabral[3], Paulo Moreira Silva Dantas[1,3]*

1 Graduate Program in Health Sciences, Federal University of Rio Grande do Norte (UFRN), Natal, RN, Brazil, 2 The Human Performance Mechanic, CUNY Lehman College, The Bronx, New York, United States of America, 3 Graduate Program in Physical Education, Federal University of Rio Grande do Norte (UFRN), Natal, RN, Brazil

☯ These authors contributed equally to this work.
* pgdantas@icloud.com

**Data Availability Statement:** All relevant data are within the paper and its Supporting Information files.

## Abstract

### Background

High frequency (1–2 times per day) low-intensity blood flow restriction (BFR) training has been recommended as a prescription approach for short durations of time to maximize relevant physiological adaptations. However, some studies demonstrate negative physiological changes after short periods of high-frequency BFR training, including prolonged strength decline and muscle fiber atrophy.

### Objectives

To provide a comprehensive overview of short-term, high-frequency blood flow restriction training, including main adaptations, myocellular stress, limitations in the literature, and future perspectives.

### Methods

A systematic search of electronic databases (Scopus, PubMed®, and Web of Science) was performed from the earliest record to April 23, 2022. Two independent reviewers selected experimental studies that analyzed physical training protocols (aerobic or resistance) of high weekly frequency (>4 days/week) and short durations (≤3 weeks).

### Results

In total, 22 studies were included in this review. The samples were composed exclusively of young predominantly male individuals. Muscle strength and hypertrophy were the main outcomes analyzed in the studies. In general, studies have demonstrated increases in strength and muscle size after short term (1–3 weeks), high-frequency low-intensity BFR training, non-failure, but not after control conditions (non-BFR; equalized training volume). Under failure conditions, some studies have demonstrated strength decline and muscle fiber atrophy

**Funding:** VSQ and PWAV were financed in part by a scholarship from the Coordenação de Aperfeiçoamento de Pessoal de Nível Superior (CAPES), Brazil - finance code 001. The funders had no role in study design, data collection and analysis, decision to publish, or preparation of the manuscript.

**Competing interests:** NR is the founder of THE BFR PROS, a BFR education company that provides BFR training workshops to fitness and rehabilitation professionals across the world using a variety of BFR devices. NR has no financial relationships with any cuff manufacturers/distributors. This does not alter our adherence to PLOS ONE policies on sharing data and materials. The other authors declare no potential or actual conflicts of interest.

after BFR conditions, accompanying increases in muscle damage markers. Significant limitations exist in the current HF-BFR literature due to large heterogeneities in methodologies.

## Conclusion

The synthesis presented indicates that short-term, high-frequency BFR training programs can generate significant neuromuscular adaptations. However, in resistance training to failure, strength declines and muscle fiber atrophy were reported. Currently, there are no studies analyzing low-frequency vs. high-frequency in short-term BFR training. Comparisons between resistance exercises of similar intensities (e.g., combined effort) are lacking, limiting conclusions on whether the effect is a product of proximity to failure or a specific effect of BFR.

## 1 Introduction

Low-load resistance training programs (20–50% 1 repetition maximum [1RM]), combined with arterial blood flow restriction (BFR) of the exercised limb, can promote strength gains and muscle hypertrophy similar to non-BFR high-load training (80% 1RM) [1, 2]. In addition, low-intensity aerobic training programs with BFR can promote muscle hypertrophy, increases in lower limb strength, and aerobic capacity despite low intensities of training, adaptations that may not be achieved with traditional non-BFR low-intensity aerobic training [3, 4]. Therefore, the technique may be useful for people with limitations to high-intensity training, including people recovering from injury, surgery, or the frail elderly.

Regardless of the type of exercise (aerobic or resistance), BFR exercise is recommended to be performed 2–3 times per week (low-frequency) when interventions lasts longer than three weeks [5]. For short-term interventions (<3 weeks), the recommended weekly frequency can be increased to 1–2 times per day (high-frequency) [5]. High-frequency low-load BFR training can be a potentially useful tool to accelerate recovery in clinical rehabilitation settings since the technique can provide positive physiological adaptations in short terms [6]. Furthermore, the inclusion of high-frequency low-load BFR resistance training blocks (5 times per week) in a traditional (non-BFR) high-load resistance training routine appears to maximize hypertrophic adaptation in well-trained powerlifting athletes [7]. These results provide evidence that short periods of high-frequency low-load training with BFR may be a strategy capable of maximizing hypertrophy in individuals with a high level of training.

The literature provides support for prescribing high-frequency BFR training in different contexts. However, there is a dire need to consider relevant physiological effects of application when using this prescription model. In a recent study, Nielsen et al. [8] identified increases in capillary basement membrane thickness after a short-term high-frequency low-load BFR resistance training program (one to two daily sessions per day with weekends off). This observed change can compromise the diffusion of myocellular oxygen and the supply of nutrients, indicating a heightened stress response to the BFR stimulus [8]. Additionally, different prescribed high-frequency training protocols can result in divergent outcomes. To illustrate, Ladlow et al. [6] identified that three weeks of high-frequency low-load BFR resistance training (9 sessions per week) in acute rehabilitation patients was able to increase the strength evaluated 24 hours after the intervention. However, Nielsen et al. [9] did not identify strength increases 5 days after three weeks of high-frequency low-load BFR resistance training (7–9 sessions per week)

in healthy untrained participants and one study in a similar population identified strength declines after a block of high-frequency low-load BFR resistance training (7 sessions per week) [10]. Regarding muscle fiber size, a study identified a reduction in the cross-sectional area of type I fiber ten days after high-frequency low-load BFR resistance training [10], while a second study identified an opposite response (i.e., hypertrophy) [11].

Considering the possibility of negative physiological changes and divergences in the results of studies on short-term high-frequency BFR training, it is important to provide an overview of this model of BFR training for professionals and researchers, aiming to guide prescription of high-frequency protocols. Therefore, this scoping review aimed to systematically synthesize the available scientific literature on short-term high-frequency BFR training, describe the protocols tested, outcomes evaluated, main findings and describe limitations in the evidence base that impact the strength of evidence. We will also propose areas within high-frequency BFR training application for future studies to investigate.

## 2 Methods

This scoping review was reported in accordance with the Preferred Reporting Items for Systematic reviews and Meta-analyses extension for Scoping Reviews (PRISMA-ScR) [12]. The final protocol of this review was prospectively registered in the Open Science Framework (OSF) on April 23, 2022 (https://osf.io/f3ksx).

### 2.1 Eligibility criteria

Experimental studies that analyzed short-term ($\leq$3 weeks) BFR training with weekly high-frequencies (>4 times/week) published between 2000 and 2022 were eligible for this review. We only included studies that evaluated humans. There was no restriction by age, sex, or clinical condition. In addition, there was no restriction by publication language. Studies that analyzed exclusively passive BFR protocols, case reports, expert opinion, and narrative or systematic reviews were not considered for the analyses.

### 2.2 Information sources

Searches were performed on April 23, 2022 in the following databases: Scopus, PubMed, Web of Science. The reference list of studies eligible for review was screened to find potential studies that were not identified in the searched databases. In addition, Google Scholar citations was used to identify studies that were eligible for review. This procedure was performed to track relevant studies on the investigated topic.

The search strategy combined the following descriptors and Boolean operators (AND/OR): ("blood flow restriction therapy" OR "BFR therapy" OR "BFR therapies" OR "blood flow restriction training" OR "blood flow restriction exercise" OR "BFR training" OR "BFR resistance training" OR "blood flow restriction resistance training" OR "blood flow restriction resistance exercise" OR "BFR aerobic training" OR "blood flow restriction aerobic training" OR "Kaatsu training" OR Kaatsu OR "high-frequency blood flow restriction" OR "high-frequency blood flow restriction training" OR "high-frequency blood flow restriction resistance training"). No additional filters or search limitations were used.

### 2.3 Selection of sources of evidence

The selection process was performed by two reviewers (VSQ, PWAV) blindly and independently. Disagreements between reviewers were resolved by a third reviewer (NR). The screening process was divided into three stages: (i) elimination of duplicates; (ii) reading of titles and

abstracts; (iii) reading of full articles. The Rayyan QCRI® (Rayyan QCRI, Qatar Computing Research Institute, HBKU, Doha, Qatar) [13] was used to eliminate duplicates and assist in the screening of titles and abstracts.

After a complete reading of the studies, two reviewers (VSQ and PWAV) extracted data from the eligible studies. The following information was extracted: sample size, study design, participant characteristics (sex, age, weight, height, training status, clinical condition), outcomes, duration and frequency of intervention, exercise (s) tested, exercise characteristics (intensity, volume, rest intervals inter-sets), limb BFR (pressure, cuff width, duration) and statistical analyses used.

## 2.4 Data synthesis

Considering that the primary objective of scoping reviews is to map the extent, scope, and nature of a given topic and the secondary objective is the synthesis of divergent results [12], no quantitative analysis was implemented in this review. Data were presented using a qualitative synthesis. When appropriate, we graphically report the magnitude of muscle strength gain and muscle hypertrophy.

# 3 Results

## 3.1 Study selection

A total of 3561 studies were identified in the databases. After elimination of duplicates, 2317 studies remained to be screened from titles and abstracts. Thirty studies were selected for full reading, of which fourteen were eligible. Six studies were identified in citation tracking and two studies were identified in Google Scholar citations for a total of 22 included studies. Details of the screening process are reported in Fig 1.

## 3.2 Participant characteristics

The studies included 360 participants (men, $n = 330$ [92%]; women, $n = 30$ [8%]). The sample size varied between 5 and 24 participants. The mean age of the participants evaluated in the studies ranged from 20 to 34 years old. Three studies (16%) evaluated athletes [7, 14, 15]; one study evaluated powerlifters [7], one study evaluated college track and field athletes (sprinters and jumpers) [14], and one study evaluated college basketball athletes [15]. Three studies (16%) evaluated injured individuals [6, 16, 17]. The remaining studies (68%) evaluated healthy untrained individuals. Details of participant characteristics are reported in Table 1.

## 3.3 Study design

The parallel design (between-subjects) was adopted in eighteen (82%) studies [4, 6–9, 14–20, 22–25, 27, 28]. Two (9%) studies [10, 26] adopted a within-subjects design and two (9%) studies were uncontrolled [11, 21].

## 3.4 Characteristics of training interventions

Low-intensity aerobic training (walking training) was investigated in five (23%) of the included studies [4, 15, 18–20]. None of the studies personalized the pressure application in training. The comparators for this type of intervention were walking without BFR (control-walk) with equalized training volume and intensity. All studies analyzed intermittent protocols (5 sets of 2–3 minutes with 1 minute of inter-set rest).

Regarding studies that investigated resistance training, two studies did not introduce comparators (control conditions) [11, 21], one study compared failure versus non-failure (both

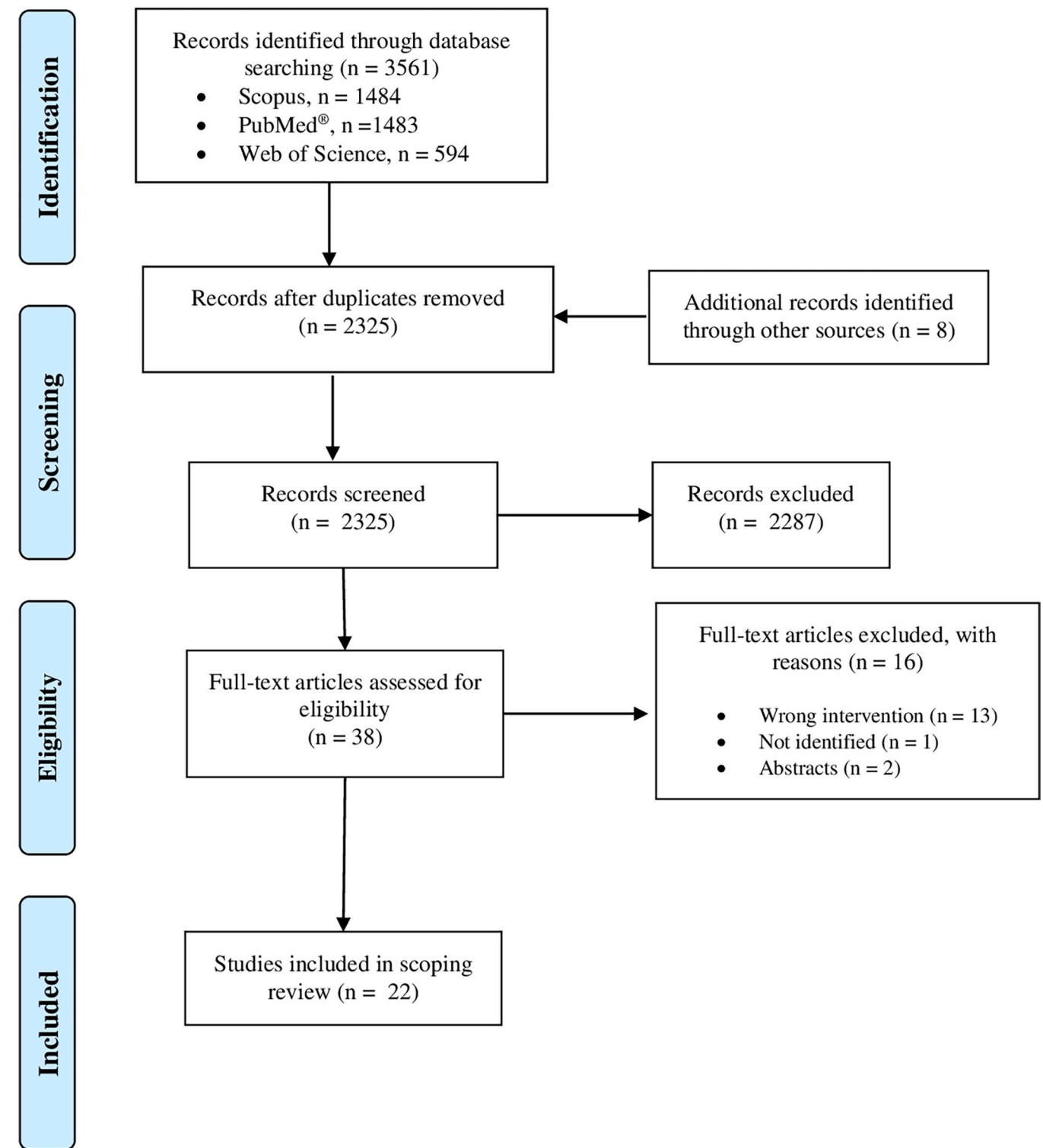

**Fig 1. PRISMA flow diagram of study selection.**

**Table 1. Characteristics of the participants.**

| Study | Sample size (n = 360) | Sex: F/M | Age (Years) | Height (cm) | Weight (kg) | RT experience |
|---|---|---|---|---|---|---|
| Abe et al. [22] | 16 | 0/16 | 23.6 ± 6.5 | 172.4 ± 6.5 | 64.3 ± 9.8 | Untrained |
| Abe et al. [14] | 15 | 0/15 | NR | 175.3 ± 5.5 | 66.8 ± 4.2 | Trained |
| Beekley et al. [18] | 18 | 0/18 | 21.3 ± 2.8 | 174.0 ± 5.0 | 64.8 ± 5.3 | Untrained |
| Yasuda et al. [23] | 05 | 0/5 | Not reported | Not reported | Not reported | Not reported |
| Abe et al. [4] | 18 | 0/18 | 21.3 ± 2.8 | 174.1 ± 4.3 | 64.8 ± 5.3 | Untrained |
| Fujita et al. [24] | 16 | 0/16 | 21.7 ± 3.0 | 171.0 ± 0.5 | 63.5 ± 6.2 | Untrained |
| Abe et al. [19] | 12 | 0/12 | Not reported | 176.0 ± 0.5 | 64.3 ± 3.3 | Untrained |
| Yasuda et al. [25] | 10 | 0/10 | Not reported | 172.0 ± 5.0 | 66.0 ± 7.0 | Untrained |
| Park et al. [15] | 14 | 0/14 | 20.4 ± 1.2 | 189.3 ± 6.4 | 88.1 ± 6.6 | Not reported |
| Sakamaki et al. [20] | 17 | 0/17 | 21.2 ± 1.9 | 174.0 ± 0.07 | 21.2 ± 1.9 | Untrained |
| Sakamaki et al. [26] | 13 | 8/5 | 26.5 ± 3.9 | 165.0 ± 0.03 | 57.9 ± 5.1 | Untrained |
| Nielsen et al. [27] | 20 | 0/20 | 22.3 ± 2.5 | 182.0 ± 7.6 | 81.2 ± 12.5 | Untrained |
| Iversen et al. [17] | 24 | 10/14 | 27.3 ± 8.3 | 177.9 ± 7.8 | 77.2 ± 10.8 | Untrained |
| Nielsen et al. [9] | 21 | 0/21 | 22.5 ± 2.5 | 182.0 ± 7.5 | 81.2 ± 12.5 | Untrained |
| Nielsen et al. [28] Study 1 | 20 | 0/20 | 22.5 ± 2.0 | 181.0 ± 6.0 | 82.0 ± 14.0 | Untrained |
| Nielsen et al. [28] Study 2 | 20 | 0/20 | 23.5 ± 2.5 | 182.0 ± 7.0 | 78.5 ± 6.5 | Untrained |
| Zargi et al. [16] | 20 | 4/16 | 34.5 ± 5.5 | Not reported | Not reported | Untrained |
| Bjørnsen et al. [7] | 17 | NR | 25.0 ± 5.5 | 176.5 ± 7.5 | 95.5 ± 16.0 | Trained |
| Ladlow et al. [6] | 28 | 0/28 | 30.5 ± 6.5 | 178.5 ± 6.5 | 90.0 ± 16.0 | Untrained |
| Bjørnsen et al. [10] | 13 | 4/9 | 24.0 ± 2.0 | 179.0 ± 8.0 | 78.0 ± 12.0 | Untrained |
| Nielsen et al. [8] | 21 | 0/21 | 22.3 ± 2.5 | 182.0 ± 7.6 | 81.2 ± 12.5 | Untrained |
| Bjørnsen et al. [21] | 13 | 4/9 | 24.0 ± 2.0 | 179.0 ± 8.0 | 78.0 ± 12.0 | Untrained |
| Bjørnsen et al. [10] | 17 | 0/17 | 25.0 ± 6.0 | 181.0 ± 12.0 | 80.0 ± 13.0 | Untrained |

RT: resistance training.

with BFR) [10], one study compared low-load BFR training versus non-exercise [26], eleven studies compared low-load resistance training with BFR versus work-matched non-BFR low-load resistance training [8, 9, 14, 16, 17, 22–28], three studies compared low-load resistance training with BFR versus non-BFR heavy load resistance training [6, 7, 28]. Four studies adopted more than one exercise in the resistance training session [6, 14, 22, 23], while the others adopted a single exercise. Only two studies evaluated upper limbs [25, 26]. One study evaluated the inclusion of high-frequency low-load BFR resistance training in a traditional heavy load resistance training routine [7] and one study evaluated the inclusion of high-frequency low-load BFR resistance training in a sprint/jumping training routine [14].

A single study personalized the applied BFR pressure (60% arterial occlusion pressure [AOP]) [6]. The other studies applied arbitrary pressures (e.g., 100 mmHg). One study used practical BFR [7]. Four studies applied two blocks of high-frequency low-load BFR resistance training interspersed per 9–10 days [7, 10, 11, 21]. Details of intervention characteristics are reported in Table 2.

### 3.5 Qualitative synthesis of the main results

**3.5.1 Muscle strength.** *1-RM test.* Muscle strength was evaluated in sixteen studies included in this review. Eleven studies used 1-RM tests to evaluated muscle strength [4, 7, 10, 14, 18, 19, 22–25]. Among the studies that evaluated the performance of 1-RM, eight studies had non-BFR training with equalized volume and intensity as comparators (e.g., work-

**Table 2. Characteristics of included studies.**

| Study (Year) | Study design | Training type (Exercise [s]) | Intervention duration (Weeks) | Weekly frequency (sessions/week) | Intensity | Sets number (Interval recovery) | Repetitions or duration | BFR pressure (Cuff size) | Outcomes | Reported statistic |
|---|---|---|---|---|---|---|---|---|---|---|
| Abe et al. [22] | Parallel (Randomized) | RT (Squat and leg curl) | 2 | 12 | LL-BFR: 20% 1RM LL: 20% 1RM | LL-BFR: 3 (30s) LL: 3 (30s) | 15 | 160–240 mmHg (NR) | Muscle strength, muscle size, IGF-1, CK, Mb, LP | p-values; no power analysis |
| Abe et al. [14] | Parallel (Randomized) | RT (Squat and leg curl) | 1 | 16 | LL-BFR: 20% 1RM LL: 20% 1RM | LL-BFR:3 (30s) LL:3 (30s) | 15 | 160–240 mmHg (NR) | Muscle strength, muscle size, sprint time, jump height | p-values; no power analysis |
| Beekley et al. [18] | Parallel (Randomized) | AT (Walking) | 3 | 12 | LI-BFR: 50m/min LI: 50m/min | LL-BFR: 5 (60s) LL: 5 (60s) | 120s | 160–230 mmHg (NR) | Muscle strength, muscle size, ALP, IGF-1 | p-values; no power analysis |
| Yasuda et al. [23] | Parallel (No randomized) | RT (Squat and leg curl) | 2 | 12 | LL-BFR: 20% 1RM LL: 20% 1RM | LL-BFR: 3 (30s) LL: 3 (30s) | 15 | 160–240 mmHg (NR) | Muscle strength, muscle size | p-value; post-hoc power analysis performed |
| Abe et al. [4] | Parallel (Randomized) | AT (Walking) | 3 | 12 | LI-BFR: 50m/min LI: 50m/min | LL-BFR: 5 (60s) LL: 5 (60s) | 120s | 160–230 mmHg (NR) | Muscle strength, muscle size, CK, Mb, IGF-1, IGFBP-3, GH, testosterone, cortisol | p-values; no power analysis |
| Fujita et al. [24] | Parallel (Randomized) | RT (Knee extension) | 1 | 12 | LL-BFR: 20% 1RM LL: 20% 1RM | LL-BFR: 4 (30s) LL: 4 (30s) | 30-15-15-15 | 200 mmHg (NR) | Muscle strength, muscle size, CK, Mb, IL-6 | p-values; no power analysis |
| Abe et al. [19] | Parallel (Randomized) | AT (Walking) | 3 | 6 | LI-BFR: 50m/min LI: 50m/min | LI-BFR: 5 (60s) LI: 5 (60s) | 120s | 160–230 mmHg (NR) | Muscle strength, muscle size | p-values; no power analysis |
| Yasuda et al. [25] | Parallel (Randomized) | RT (Bench press) | 2 | 12 | LL-BFR: 30% 1RM LL: 30% 1RM | LL-BFR: 4 (30s) LL: 4 (30s) | 30-15-15-15 | 100–170 mmHg (NR) | Muscle size, muscle strength, IGF-1, IGFBP3, CK, Mb | p-values; no power analysis |
| Park et al. [15] | Parallel (Randomized) | AT (Walking) | 2 | 12 | LI-BFR: 4km/h LI: 4km/h | LL-BFR: 5 (60s) LL: 5 (60s) | 180s | 160–230 mmHg (11 cm) | Muscle strength, cardiorespiratory endurance, anaerobic power, BF%, BP, HR, SV, CO, RPE | p-values; no power analysis |
| Sakamaki et al. [20] | Parallel (Randomized) | AT (Walking) | 3 | 12 | LI-BFR: 50m/min LI: 50m/min | LL-BFR: 5 (60s) LL: 5 (60s) | 120s | 160–230 mmHg (NR) | Muscle size | p-values; no power analysis |
| Sakamaki et al. [26] | Crossover/within-subject (Randomized) | RT (Elbow flexion) | 1 | 6 | LL-BFR: 30% 1RM LL: 30% 1RM | LL-BFR: 4 (30s) LL: 4 (30s) | 30-15-15-15 | 80–100 mmHg (3 cm) | Muscle size, muscle strength, estradiol, progesterone and testosterone | p-values; no power analysis |
| Nielsen et al. [27] | Parallel (No randomized) | RT (Knee extension) | 3 | 7-8-9 | LL-BFR: 20% 1RM LL: 20% 1RM | LL-BFR: 4 (30s) LL: 4 (30s) | LL-BFR: Failure LL: Work-matched to LL-BFR | 100 mmHg (15 cm) | Muscle strength, muscle size, myonuclei, SC | p-values; no power analysis |
| Iversen et al. [17] | Parallel (No randomized) | RT (Knee extension) | 2 | 12 | LL-BFR: non-resistance LL: non-resistance | LL-BFR: 5 (180s) LL: 5 (180s) | 20 | 130–180 mmHg (14 cm) | Muscle size | p-values; no power analysis |
| Nielsen et al. [9] | Parallel (No randomized) | RT (Knee extension) | 3 | 7-7-9 | LL-BFR: 20% 1RM LL: 20% 1RM | LL-BFR: 4 (30s) LL: 4 (30s) | LL-BFR: Failure LL: Work-matched to LL-BFR | 100 mmHg (13.5 cm) | Muscle strength, muscle size, pain, annexin A6, CaMKII, SNO-CYS. | p-values; no power analysis |

(*Continued*)

**Table 2.** (Continued)

| Study (Year) | Study design | Training type (Exercise [s]) | Intervention duration (Weeks) | Weekly frequency (sessions/week) | Intensity | Sets number (Interval recovery) | Repetitions or duration | BFR pressure (Cuff size) | Outcomes | Reported statistic |
|---|---|---|---|---|---|---|---|---|---|---|
| Nielsen et al. [28] Study 1 | Parallel (Randomized) | RT (Knee extension) | 1 | 7 | LL-BFR:20% 1RM HL: 70% 1RM | LL-BFR: 4 (30s) HL: 4 (90s) | Failure | 100 mmHg (13.5 cm) | CK, inflammatory response, oxidative capacity, DOMS, pain. | p-values; no power analysis |
| Nielsen et al. [28] Study 2 | Parallel (No randomized) | RT (Knee extension) | 3 | 7-7-9 | LL-BFR: 20% 1RM LL: 20% 1RM | LL-BFR: 4 (30s) LL: 4 (30s) | LL-BFR: Failure LL: Work-matched to LL-BFR | 100 mmHg (13.5 cm) | HSP, inflammatory response, tenascin C, central nuclei | p-values; no power analysis |
| Zargi et al. [16] | Parallel (quasi-randomized) | RT (Knee extension) | 1 | 5 | LL-BFR: 40 RM LL: 40 RM | LL-BFR: 6 (45-90s) LL: 6 (45-90s) | LL-BFR: Failure LL: Work-matched to LL-BFR | 150 mmHg (14 cm) | Muscle strength, muscle endurance, EMGs, O₂Hb, HHb, BF$_m$ | p value; a priori power analysis performed |
| Bjørnsen et al. [7] | Parallel (Randomized) | RT (Squat) | 2 (Two blocks) | 5 | LL-BFR: 24% and 31% 1RM LL: 74% and 76% 1RM | LL-BFR: 4 (30s) LL: 6–7 (30s) | BFR: 30-15-12-8 HL: 1–6 | Practical (7.6 cm) | Muscle strength, muscle size, SC, myonuclei ribosomal capacity, capillarization | p-value and CI95%; a priori power analysis performed |
| Ladlow et al. [6] | Parallel (Randomized) | RT (BFR: Leg press, knee extension; HL: deadlift, back squat, lunges) | 3 | 9 | LL-BFR: 30% 1RM HL: 6–8 RM | LL-BFR: 4 (30s) HL: 4 (180s) | BFR: 30-15-15-15 HL: 6–8 | 60% AOP (10 cm) | Muscle size, muscle strength, endurance, balance, pain | p-values; no power analysis |
| Bjørnsen et al. [10] | No controlled | RT (Knee extension) | 2 (Two blocks) | 7 | LL-BFR: 20% 1RM | LL-BFR: 4 (30s) | Failure | M: 100 mmHg F: 90 mmHg (14.5 cm) | Muscle strength, muscle size, CK, Mb, DOMS, RPE, pain, myonuclei, Pax7 mRNA, p21 mRNA, MyoD mRNA, myogenin mRNA, Cyclin D1 mRNA, Cyclin D2 mRNA, Myostatin mRNA, IGF1R mRNA. | p-value and CI95%; a priori power analysis performed |
| Nielsen et al. [8] | Parallel (No randomized) | RT (Knee extension) | 3 | 7–9 | LL-BFR: 20% 1RM LL: 20% 1RM | LL-BFR: 4 (30s) LL: 4 (30s) | LL-BFR: Failure LL: Work-matched to LL-BFR | 100 mmHg (13.5 cm) | Capillarization, perivascular basal membrane | p-values; no power analysis |
| Bjørnsen et al. [21] | No controlled | RT (Knee extension) | 2 (Two blocks) | 7 | LL-BFR: 20% 1RM | LL-BFR: 4 (30s) | Failure | M: 100 mmHg F: 90 mmHg (14.5 cm) | CK, Mb, DOMS, RPE, pain, HSP, immune response, inflammatory response, glycogen contente | p-values; no power analysis |

(*Continued*)

**Table 2.** (Continued)

| Study (Year) | Study design | Training type (Exercise [s]) | Intervention duration (Weeks) | Weekly frequency (sessions/week) | Intensity | Sets number (Interval recovery) | Repetitions or duration | BFR pressure (Cuff size) | Outcomes | Reported statistic |
|---|---|---|---|---|---|---|---|---|---|---|
| Bjørnsen et al. [10] | Crossover/within-subject (Randomized) | RT (Knee extension) | 2 (Two blocks) | 7 | LL-BFR: 20% 1RM | LL-BFR: 20% 1RM (30s) | Failure 30-15-15-15 | 100 mmHg (14.5 cm) | Muscle size, muscle strength, myonuclei, CS, echo intensity, DOMS, pain, RPE, CK, Mb | p-value and CI95%; a priori power analysis performed |

1RM: 1 repetition maximum; ALP: alkaline phosphatase; AOP: arterial occlusion pressure; AT: aerobic training; BF, body fat; $BF_m$, muscle BF; BFR: blood flow restriction; BP, blood pressure; CaMKII, Ca2+/calmodulin–dependent protein kinase II; CI95%– 95% Confidence Intervals; CK: creatine kinase; CO: cardiac output; CON: control; CSA: cross–sectional area; DOMS: delayed onset muscle soreness; GH: growth hormone; HHb, deoxygenated hemoglobin; HL: high–load; HR: heart rate; HSP: heat shock protein; IGF–1: insulin–like growth factor I; IGFBP–3: IGF–binding protein; IL–6: interleukin 6; LL: low–load; LL–BFR: low–load with blood flow restriction; LP: lipid peroxide; mRNA, messenger RNA; Mb: myoglobin; NR: not reported; Pax7, paired box 7; $O_2Hb$, oxygenated hemoglobin; RPE: rate of perceived exertion; RT: resistance training; sEMG, surface electromyography; SC: satellite cells; SNO–CYS, S–Nitroso–Cysteine; SV–stroke volume.

matched). Except for a single study [23] that did not include analyses for this outcome, all studies reported a significant increase in 1-RM performance after training with BFR (low-load). Only one study identified an increase in 1-RM performance after the comparator (control condition) [22], but the magnitude was lower than in the BFR training condition (See Fig 2A). One study evaluated 1-RM performance after two blocks of high-frequency low-load BFR resistance training separated by 10-days of rest, under failure vs. non-failure repetition schemes; this study evidenced a decline in 1-RM performance after the first training block in the failure condition while no change was evidenced in the non-failure condition [10]; an increase in 1RM performance was observed 17 and 24 days after the second training block in both conditions. Another study from the same group showed a significant decline in 1-RM performance after a block of high-frequency low-load BFR resistance training that took 20 days after the second block of training to induce strength supercompensation [11]. One study evaluated two blocks of high-frequency low-load BFR resistance training vs. non-BFR heavy load training in a traditional high-load training routine in well-trained athletes; as a result, it was evidenced that supplementation with heavy load training was able to significantly increase

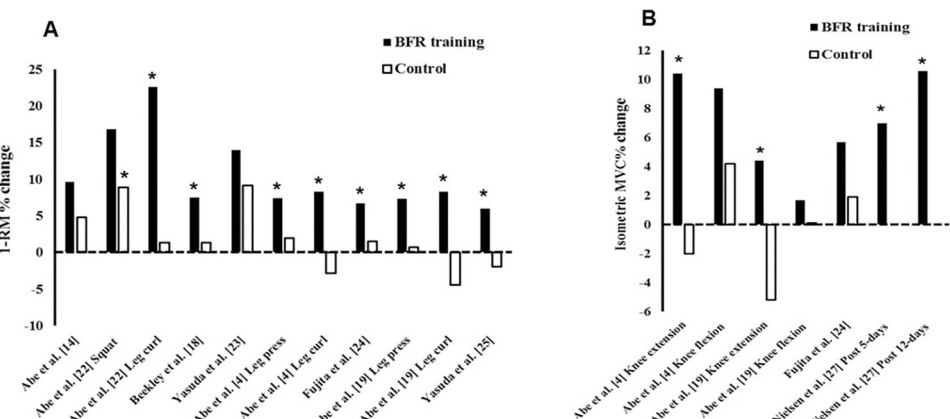

**Fig 2. Percentage (%) of muscle strength gain reported in BFR training vs. non–BFR training (control).** Note: *, significance; 1RM, repetition max; BFR, blood flow restriction; MVC, maximum voluntary.

1-RM performance, while supplementation with high-frequency BFR training did not alter strength [7].

*5-RM test*. One study used a 5-RM test as a measure of lower limb muscle strength [6]. Ladlow et al. [6] analyzed the effect of high-frequency low-load BFR resistance training versus non-BFR high-load training on 5-RM test performance in leg press and knee extension. Both conditions significantly increased 5-RM performance with no difference between conditions.

*Maximal voluntary contraction (MVC)*. Isometric maximal voluntary contraction (MVC) was evaluated in seven studies [4, 6, 16, 19, 24, 26, 27], and isokinetic MVC in three studies [7, 9, 15]. Four studies analyzed isometric MVC in low-load BFR training and non-BFR training [4, 19, 24, 27]. All studies included analyses of the isometric strength of the knee extensors and only one did not identify a significant increase after BFR training [24], while no study identified increases in isometric strength after non-BFR training (low-intensity/load). Two studies analyzed isometric knee flexor strength; however, no study reported a significant increase with- or without-BFR (See Fig 2B). One study compared the effect of high-frequency low-load BFR training versus heavy load non-BFR training on isometric strength of the hip extensors [6]; the study in question identified an increase after high-frequency low-load BFR training, but not after high-load training without BFR. One study evaluated the effect of high-frequency low-load BFR resistance training versus control (no exercise) on isometric elbow flexor MVC in men and women in the luteal and follicular phases [26]. The study in question did not identify changes in the comparator condition, while strength increases were reported after high-frequency low-load BFR resistance training in men and women, but only in the luteal phase.

Regarding isokinetic MVC, one study identified an increase in slow ($30˚s^1$) velocity isokinetic MVC 12 days after high-frequency low-load BFR resistance training, but not after 5 days [9]. For fast ($240˚s^1$) velocity isokinetic MVC, there was a decline 5 days after high-frequency low-load BFR training. No changes were reported in the comparator group (low-load non-BFR training). One study identified increased isokinetic MVC of knee flexors, but not knee extensors, after high-frequency low-intensity aerobic training with or without BFR, however, the relative changes were significantly greater in BFR high-frequency aerobic training [15]. One study [6] evaluated the inclusion of two blocks of high-frequency low-load BFR resistance training versus non-BFR heavy load training in a traditional high-load training routine; as a result, it was observed that supplementation with high-frequency BFR training significantly improved knee extensor isokinetic MVC performance at $60˚s^1$ while supplementation with heavy load training did not.

**3.5.2 Muscle endurance.** Muscle endurance was evaluated in one study [16]. Zargi et al. [16] evaluated sustained submaximal (30% isometric MVC) isometric contraction time in pre-surgical anterior cruciate ligament reconstruction patients. After 4 weeks post-surgery, when all participants were involved in a rehabilitation program (without BFR training), the high-frequency BFR training group exhibited ~50% greater sub-maximal quadriceps endurance than the non-BFR work matched comparator group, but the differences between groups washed out after 12 weeks.

**3.5.3 Jump and sprint performance.** One study evaluated jump and sprint performance. Abe et al. [14] used electronic timing system to assess the time of a 30-meter sprint. Three different jump tests (standing jump, standing triple jump, and standing 5-step jump) were performed using a long-jump pit. For all measurements, the authors performed three assessments and adopted the best performance. The results of this study indicate low-load BFR resistance training produced decreases in 30-m sprint time that was attributed to improvements in early acceleration during the first 10-m, while no change was detected in the comparator condition (non-BFR low-load resistance training). However, no improvements in jumping performance were noted in either condition.

**3.5.4 Cardiorespiratory endurance.** A single study [15] analyzed cardiorespiratory endurance. This study showed that maximal oxygen volume (VO$_2$max [ml/min/kg]) significantly increased after two weeks of high-frequency low-intensity BFR aerobic training (4 km/h interval walking), while no differences were reported in the work-matched non-BFR training comparator condition.

**3.5.5 Anaerobic power.** A single study [15] evaluated anaerobic capacity. This study showed that power significantly increased after two weeks of high-frequency low-intensity BFR aerobic training (4 km/h interval walking), while no differences were reported in the work-matched comparator condition (high-frequency non-BFR training).

**3.5.6 Muscle size.** *Macroscopic measurements*. Nine studies used magnetic resonance imaging to assess hypertrophic adaptations (Cross-sectional area [CSA] and muscle volume) from high-frequency low-load BFR training [4, 6, 11, 17–20, 24, 26]. Muscle thickness obtained by ultrasound was used in five studies [7, 10, 11, 14, 25]. Iversen et al. [17] analyzed the quadriceps CSA before and 16 days after anterior cruciate ligament reconstruction. From the 2nd to the 14th post-surgery day, a group of patients underwent an BFR exercise intervention (without external load), while a comparator group underwent an exercise intervention without BFR (without external load). The two groups experienced similar declines in quadriceps CSA (-13.8% vs. -13.1% for BFR and comparator, respectively; p = 0.626). One study [6] compared quadriceps CSA and thigh volume before and after three weeks of high-frequency low-load BFR training vs. traditional heavy load training (3 times per week). The two training models induced a significant increase in the outcomes of interest, without differences between the interventions. Nine studies analyzed CSA/volume and muscle thickness in high-frequency low-load training vs. non-BFR training with equal training volume and intensity. Overall, these studies found that the BFR condition promoted a significant increase in muscle size, while the comparator condition did not significantly alter this variable (See Fig 3). One study evaluated two blocks of high-frequency low-load BFR training versus heavy load training

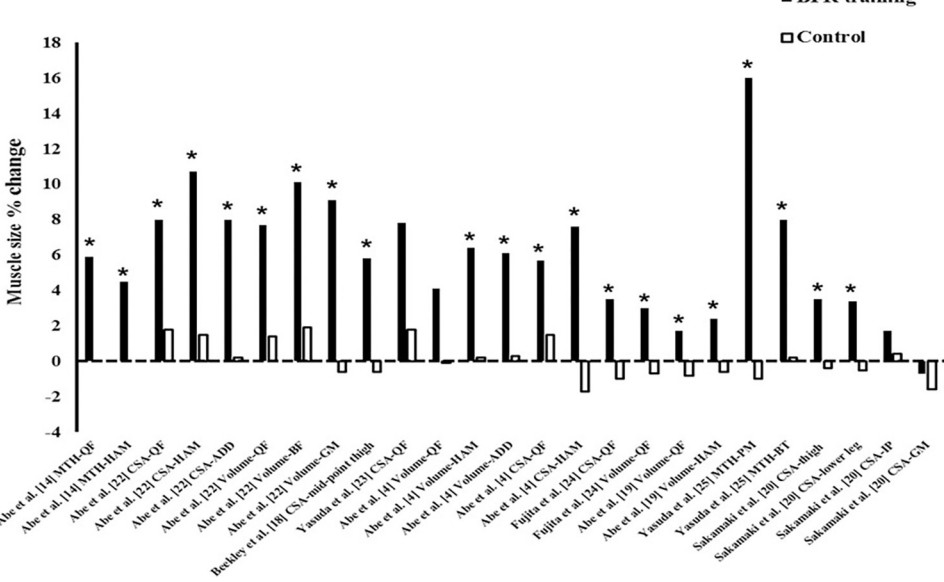

**Fig 3. Percentage (%) of muscle size gain reported in BFR training vs. non–BFR training (control).** Note: *, significance from baseline (p < 0.05); ADD, adductors; BF, biceps femoris; BFR, blood flow restriction; CSA, cross–sectional area; GM, gluteus maximus; HAM, hamstrings; IP, iliopsoas; MTH, muscle thickness; QF, quadriceps femoris.

without BFR in a traditional high-load training routine; as a result, it was shown that supplementation with high-frequency low-load BFR resistance training was able to significantly increase the thickness of the quadriceps femoris muscles, while supplementation with heavy load training did not [7]. One study [10] evaluated muscle size after two blocks of high-frequency low-load BFR training separated by 10-days of rest, under failure vs. non-failure exercise protocols; evaluations were made every two days within the training blocks, in the recovery week, and following 3-, 10-, 17-, 24- days after the last training block. The study concluded no significant muscle size differences between conditions. Increases in the rectus femoris and vastus lateralis were observed at all time points evaluated, while for the vastus intermedius, increases were not observed during the rest week and 10-, 17- and 24-days post-training. One study [11] looked at a similar training protocol; the study in question identified an increase in the size of the rectus femoris on the fifth day of the first and second training block and 3, 5 and 10 days after the last training block; for the vastus lateralis, increases were evident on the fourth and fifth days of the first training block, in the recovery week, in all assessments performed in the second training block and 3, 5 and 10 days after the second training block period.

*Microscopic measurements*. Five studies analyzed cross-sectional area (CSA) of muscle fibers. One study [10] compared two blocks of high-frequency low-load BFR training, interspersed with ten days of rest, under failure vs. non-failure repetition protocols. Analyses were performed at baseline, during the rest week and ten days after the last block. In the failure condition, there was a significant decrease in type I muscle fiber CSA ten days after the second training block. One study [11] tested the same training protocol and showed the opposite response, that is, an increase in CSA of type I fibers ten days after the first training block, but there was a decrease in CSA observed after four days of training. In addition, a decrease in CSA of type II fibers was reported after four days of training in the recovery week. One study [7] evaluated two blocks of high-frequency low-load BFR resistance training versus heavy load training without BFR in a traditional high-load training routine; in the BFR condition, there was a significant increase in CSA of type I fibers, but not of type II fibers; in the comparator condition (heavy load training), no significant differences were reported. One study [23] showed a significant increase in CSA of type II fibers after high-frequency low-load BFR training with no difference in type I fibers CSA; there were no differences in the CSA of type I or II fibers in the comparator condition (non-BFR exercise). One study [9] analyzed muscle fiber CSA in high-frequency low-load training with and without BFR. Assessments were made at baseline, after eight days of training, and after three and ten days of detraining; Increases were evidenced at all time points in the BFR condition, while in the comparator condition, increases were evidenced only after eight days of training.

**3.5.7 Satellite cells.** Four studies analyzed satellite cells (SC) using paired box factor 7 (Pax-7) or Neural Cell Adhesion Molecule (NCAM) analysis. One study [27] evaluated Pax-7 marker at baseline, after eight days of high-frequency low-load training with and without BFR, and three and ten days after the interventions ended. In relation to baseline values, Pax-7 per myofibril, myofiber CSA, and myonuclei of the type I and II increased significantly at all time points analyzed in the BFR condition but did not change in the work-matched comparator condition. One study found that the addition of two blocks of high-frequency low-load BFR training or traditional heavy load training to a traditional high-load training routine did not generate changes in Pax-7 content per muscle fiber in nationally qualified powerlifters [7]. One study [11] evaluated two blocks of high-frequency low-load BFR interspersed for 10 days of rest. Assessments of Pax-7 content per muscle fiber were performed on the fourth day of the first training block, in the rest week, and three and ten days after the second training block. For type I and II fibers, increases in Pax-7 content per muscle fiber were evidenced at all time

points analyzed. One study [10] evaluated SC after two blocks of high-frequency low-load BFR training separated by 10 days of rest, under failure vs. non-failure repetition protocols; assessments were made at baseline, in the rest week, and ten days after the second training block; increases in NCAM content per muscle fiber were evidenced at all time points.

**3.5.8 Myonuclei.** One study [28] analyzed central nuclei in high-frequency low-load training to failure with and without BFR (work-matched to BFR condition) at baseline, after eight days of training, and three and ten days after completion of the training program. In relation to baseline values, there was a significant increase in the number of central nuclei in type II fibers at all time points evaluated in the BFR condition but no changes in the comparator condition. Similarly, one study [27] analyzed myonuclei content per muscle fiber at the same time points; as a result, an increase was evidenced in type I and II fibers in the BFR condition with no changes in the comparator condition. One study [7] evaluated two blocks of high-frequency low-load BFR training vs. non-BFR heavy load training in a traditional high-load training routine; this study did not identify changes in the content of myonuclei in type II fibers but saw increases in type I fibers. No changes were evidenced in the comparator intervention (high-load). One study evaluated two blocks of high-frequency low-load BFR training separated by 10-days of rest, under failure vs. non-failure repetition protocols; Measurements of myonuclear content per muscle fiber were performed before the intervention, during the rest week, and ten days after the end of the intervention. Increases in myonuclei content by type I and type II fibers was evidenced ten days after the end of the intervention in the two conditions tested. In the recovery week, the no-fail condition did not significantly increase myonuclei content. Similarly, one study [11] identified increases in myonuclei content per muscle fiber type I and II after 3 and 10 days a block of high-frequency low-load BFR training, but no increases were observed in the rest week.

**3.5.9 Muscle damage indirect markers.** Eight studies analyzed muscle damage markers [4, 10, 11, 21, 22, 24, 25, 28]. Serum creatine kinase (CK) activity was analyzed in all studies. Seven studies included analyses of myoglobin (Mb) [4, 10, 11, 21, 22, 24, 25]. Delayed onset muscle soreness (DOMS) was evaluated in four studies [10, 11, 21, 28].

Two studies analyzed serum CK activity before and after (24–48 h post-session) the first training session [24, 28]. None of the studies identified changes in CK after BFR. Four studies analyzed serum CK activity after the last training session performed [4, 22, 25, 28]. None of the studies in question identified changes in serum CK activity. Three studies [10, 11, 21] analyzed blocks of high-frequency low-load BFR training separated by a recovery period. In two of these studies [11, 21], serum CK activity was analyzed on all training days (fasted), in the recovery week, and after the first training session of each block (1 and 3 hours later). CK increased significantly in the first training block, peaking on the fifth training day and returning to baseline values by the recovery week. No significant increases were reported in the second training block. One study [10] analyzed serum CK activity at baseline, on the fourth day of training, and 2–4 hours after the first training session of the first block. Relative to baseline values, CK significantly increased at all time points analyzed. In all the studies evaluated, Mb presented similar behavior to CK.

Regarding DOMS, three studies [10, 11, 21] evaluated the variable before training sessions. These studies evaluated two blocks of high-frequency low-load BFR training separated by 10 days of recovery. One study analyzed DOMS before, 24 and 48 hours after the first and last session [28]. The latter had non-BFR high-load training as a comparator. DOMS increased significantly 24–48 hours after the first session of BFR exercise and non-BFR heavy load training, with higher scores being reported in BFR training group; a similar response was reported after the last session, but only in assessments made 24-hours post-exercise and with reduced magnitude. Regarding the articles that analyzed DOMS daily, two studies [11, 21] identified a

maximum increase on the third training day (block 1), but the values returned to baseline in the recovery week and did not increase significantly in the second training block. Finally, one study looked at DOMS in failure vs. non-failure; the authors identified that DOMS was higher in the failure condition. Furthermore, DOMS was higher in the first training block than in the second training block.

**3.5.10 Myocellular stress.** Cellular stress markers were included in three studies. Two studies analyzed three weeks of high-frequency low-load training with and without BFR [9, 28]. In both studies, assessments were made at baseline, after eight days of training, and three and ten days after discontinuation of training programs. One study [9] looked at $Ca^{2+}$/calmodulin-dependent kinase II (CaMKII), annexin A6 and S-nitroso-cysteine (SNO-CYS), while the second study [28] looked at heat shock proteins (HSP). In the first study [9], for the BFR condition, increases in CaMKII were evidenced after eight days of training and after the detraining period, while no differences were reported in the comparator condition. The other markers were not significantly altered in the BFR condition, while SNO-CYS was increased eight days after non-BFR training. In the second study [28], an increase in HSP-27 expression was evidenced after eight days of high-frequency low-load BFR training, while in the comparator condition this increase was evidenced after three days of detraining; HSP-70 was not altered in BFR condition but increased three days after in the comparator condition. In relation to the expression of intracellular and membrane HSP-27, increases were reported after eight days of BFR training, but no significant differences were reported in the comparator condition.

One study [11] evaluated two blocks of high-frequency low-load BFR training, interspersed with ten days of recovery; HSP measurements were made at baseline, on the first day of each training block, on the fourth day of the first training block, and after 3 and 10 days of detraining. Cytosolic α-B-crystallin levels were reduced on the first day of each block, the fourth day of the first training block, and the recovery week. The α-B-crystallin cytoskeletal levels increased significantly after the first session of each training block. Soluble α-B-crystallin levels were reduced after the first training session, increased in the recovery week and after ten days of detraining. There was an increase in nuclear α-B-crystallin levels at all time points, except for measurements taken after ten days of detraining. Levels of cytosolic HSP-70 increased only after ten days of detraining. HSP-70 cytoskeletal levels increased after the fourth day and after the third day of detraining. Soluble HSP-70 levels increased from the fourth day of training and remained elevated until ten days after the training program. HSP-70 nuclear levels increased following three and ten days after a detraining period. Regarding the analyzes by fiber type, it was found that higher levels of α-B-crystallin were identified in type I fibers after the first session of each training block and on the fourth day of the first training block. This response was evidenced for HSP-70, but significance was reached on the fourth day of training, in the recovery week and after three days of detraining.

**3.5.11 Inflammatory responses.** Two studies analyzed the serum concentration of interleukin 6 (IL-6) before and after the first training session [24, 28]. One of these studies used low-load training without BFR as a comparator [24], while the other had high-load training [28]. None of the studies identified significant changes in plasma IL-6 concentrations after low-load training, with or without BFR. Twenty-four hours after high-load training, there was an observed increase in IL-6 concentration (+68%). One study analyzed serum concentrations after the last session of high-frequency low-load BFR training and heavy load training [28]. An 18% decline was observed in the BFR condition (post 180-minutes), while there was no change in high-load training. Serum concentration of the tumor necrosis factor α (TNF-α) and monocyte chemotactic protein 1 (MCP-1) were analyzed in one study [28]. The study in question had heavy load training as a comparator. MCP-1 was reduced 24-hours after the first BFR training session. A significant reduction was observed between the baseline values of the first

and last BFR training session [28]. TNF-α was not altered after BFR training but was increased 180 minutes and 24 hours after the first heavy load training session.

One study [28] analyzed macrophage content by immunofluorescence before, after eight days, and three and ten days after three weeks of high-frequency training with and without low-load BFR. An increase in CD68+/CD206− content was evidenced after both conditions. CD68+/CD206+ was increased in BFR training, but not in non-BFR training. The content of CD68−/CD206+ expressed per square millimeter of fiber cross-sectional area was significantly increased after eight days of BFR training but did not change after the end of the intervention; increases were evidenced after the comparator condition. The CD68−/CD206+ content per 100 myofibers was increased in measurements taken after eight days of BFR training. This response was observed three days after the end of the intervention in both conditions.

One study [21] analyzed mRNAs of the interleukin-8 (IL-8), IL-6, interleukin-4 (IL-4), interleukin-1b (IL-1b), cluster of differentiation 163 (CD163), cluster of differentiation 68 (CD68), cyclooxygenase 2 (CoX2)/prostaglandin-endoperoxide synthase 2 (PTGS2), and TNF-α. Furthermore, CD68+ macrophages and CD66b+ neutrophils per fiber was evaluated. Biopsies were performed at baseline, after the first training day of each training block, in the recovery week and after 3, 10 and 20 days of the last training block. In relation to the baseline values, there was an increase in the expression of the mRNAs IL-6, IL-8, IL-1b, CD68, CoX2, TNF-α after the first training session of each block. IL-8 mRNAs were increased after the fourth day of the first training block. mRNAs of IL-6, IL-8, IL-1b, IL-4, CD68, CD163, TNF-α mRNAs increased three days after the last training session. After 10 days, increases were identified for mRNAs of IL-4, CD68, and CD163. CD66 per fiber was not changed at any point in time. CD68 per fiber was increased after the first day of each training block and after 3 and 10 days of the last training block.

**3.5.12 Vascular adaptations.** One study [8] analyzed the number of capillaries per muscle fiber and changes in the perivascular basement membrane in high-frequency training with and without low-load BFR. Assessments were performed at baseline, after eight days of training, and after three and ten days of detraining. Significant increases were evidenced at all time points analyzed in the BFR condition, but not in the comparator condition. Relative to baseline, capillary area was increased at all time points in the BFR condition, but not in the comparator condition. Regarding perivascular basement membrane analyses, small (n = 4), moderate (n = 1) and high (n = 1) increases were reported three days after the BFR condition, but not in the comparator condition; After 10 days, it was still possible to identify small (n = 3) and moderate (2) increases, while no changes were reported.

One study [7] evaluated at high-frequency BFR training supplementation vs. heavy load training in a traditional high-load training routine in the capillarization of type I and II fibers; an increase in the number of capillaries in type I fibers was reported in the BFR condition, while there were no differences in the control condition; no significant differences were reported in the control condition.

**3.5.13 Serious side effects.** A study [11] reported one of the participants ceased BFR due to severe pain, weakness in the quadriceps and difficulty in locomotion. The participant was unable to walk without crutches after initial sessions of high-frequency low-load BFR resistance training (CK values = 4188U/I vs. 194U/I at baseline).

## 4 Discussion

The aims of this scoping review were (i) to characterize and describe the main methodological features of high-frequency (>4 sessions/week) blood flow restriction (BFR) training in both resistance and aerobic exercise, (ii) evaluate the main outcomes across the included studies,

(iii) identify limitations in the literature and (iv) propose areas for future research on this topic. The results allowed for a comprehensive qualitative synthesis of the main methodological approaches to high-frequency BFR application. In the following paragraphs, our findings, and the implications on current and future direction of this application approach are discussed.

## 4.1 General findings identified in high-frequency BFR application across exercise modes

Despite significant heterogeneities in methodologies and applied BFR prescription factors, the current body of evidence on high-frequency blood flow restriction (HF-BFR) training indicates that BFR outperforms low-intensity non-BFR load/intensity matched exercise in almost every relevant marker tracked in this scoping review and produced equivocal results when low-load BFR resistance exercise was compared to heavy load strength training. For the purposes of brevity, we will focus on the major general findings of muscle strength and muscle mass across both resistance and aerobic exercise and finish with other observations that were compiled from the included studies.

Regardless of exercise mode, sessions per week ($> 4$), and method of assessment (e.g., maximal isometric voluntary contraction), it appears that HF-BFR training generally produces superior increases in muscle strength when compared to the same intensity exercise performed without BFR. When grouped together (BFR resistance and aerobic exercise trials) and without restricting type of strength assessed (e.g., isometric vs. isotonic), strength increases following HF-BFR were shown to be between +1.7%—+23% compared to -5%—+11.5% in the control condition (Fig 2). Except for 4 post-strength assessments [4, 19, 23, 24] statistical significance beyond the comparison (non-BFR low-intensity training) group was achieved in 78% (n = 14) of the included trials (n = 18 strength assessments). The results are in line with other reviews investigating longer-term effects ($> 3$ weeks) of BFR training on muscle strength in mostly older adults [29], and those with knee injuries [30]. However, as this review was not a meta-analysis and relied on qualitative syntheses, caution is warranted in making firmer conclusions regarding the potential of HF-BFR to induce similar strength gains as longer duration protocols. It should be noted that the scientific literature provides a limited number of comparisons between high-frequency, low-load BFR training versus high-load training (non-BFR). In long-term interventions ($>3$ weeks) for muscle strength, the results tend to favor high-load training [31]. Ladlow et al. [6] did not identify this superiority when comparing high-frequency, low-load BFR training versus high-load training (non-BFR). In this context, it is possible that the difference between interventions can only be observed in long-term interventions.

Similar to the positive effects of HF-BFR on strength outcomes, muscle size changes were also evident in favor of the BFR condition, irrespective of exercise mode. Muscle size changes post-HF-BFR were evidenced to be between -0.6%—+16% while the comparator group evidenced changes between -1.7%—+2.4% (Fig 3). Prior reviews on muscle size changes following longer duration protocols ($> 3$ weeks) with lesser weekly frequencies ($< 4$) show a similar trend in a variety of populations [29, 31, 32].

The current body of evidence on HF-BFR protocols in both aerobic and resistance training appear to support that similar muscle size changes can be achieved in a shorter duration of time compared to longer duration protocols. Like the above section on muscle strength, caution is warranted when extrapolating the findings to practice due to the significant heterogeneities in the protocols. It is likely that the combination of higher frequencies of training using a more fatiguing stimulus (BFR) with minimal to no muscle damage marker accretion enables the hypertrophic process to occur at a larger rate compared to the same exercise without BFR.

In a relatively high-frequency BFR intervention (4x/week), Shiromaru et al. [33] compared the time course of adaptations of a 12 sessions of non-failure BFR (30% 1-RM) performed over 3 weeks to the same number of sessions performed over 6 weeks (2x/week) in high-load (80% 1-RM) leg extension training. Their results seemed to indicate that the hypertrophic response to BFR occurred without the presence of edema (identified from the fast recovery of spin echo inversion [FSE-STIR] obtained through magnetic resonance imaging in 3 weeks) whereas in high-load training, concomitant edema occurs (at week 3), impairing the initial adaptation process yet over the same number of sessions, tended to induce greater muscle hypertrophy and muscle strength (at 6 weeks). This lends credence to the idea that a HF-BFR protocol induces a low level of muscle damage/edema allowing a higher volume and/or frequency of training to be performed without detrimental impact on muscular hypertrophy. Yet it is important to highlight that the BFR protocol consisted of 3 sets of 15, a volume typically lower than what is recommended in practice. Nonetheless, we speculate that a similar outcome occurs during non-failure HF-BFR protocols, partially explaining the muscular benefits observed.

The current review also investigated cellular changes including myofiber hypertrophy, satellite cell proliferation, myocellular stress, indirect muscle marker production, inflammatory responses, and vascular adaptations to HF-BFR. Knowledge of these responses to HF-BFR is important because it can help shape the safety profile of the exercise prescription. Highlights from our analyses revealed the following: (1): Heterogeneous results were evidenced for muscle fiber CSA analyses; some studies showed preferential hypertrophy of type I fibers [7, 11], while one study reported a decrease in CSA of this fiber type [10]. This is a curious aspect, since two of these studies [10, 11] tested the same protocol (two blocks of the HF-BFR (7 session/week), with 10-days of rest) and the measurements were taken at the same time point (10 days post). The researchers responsible for both studies justify that this divergence can be, in part, explained by the level of training of the volunteers and the administration of post-exercise protein supplementation present in only one of the interventions. Finally, one study reported a preferential increase in type II fibers; (2): HF-BFR does not appear to induce significant production of indirect markers of muscle damage evidenced by post-exercise assessment of CK and Mb in non-failure configuration; studies that investigated failure protocols demonstrated increases in serum CK and Mb activity [10, 11, 21]; (3): Satellite cell proliferation and markers of positive myofiber adaptations appear to be above what occurs in the same exercise performed without BFR; (4): Myocellular stress occurs to a greater degree in the BFR condition compared to the control condition, but the response appears to be attenuated over time; (5): Inflammatory blood markers (e.g., Interleukin-6) appear to be minimal post-exercise in the HF-BFR; (6): HF-BFR may induce signs of vascular stress as evidenced by perivascular thickening. Muscle damage has been postulated as a potential mechanism of satellite cell activation (See Fig 4). Therefore, the analyzed markers are directly linked.

While in-depth discussion of each observation is likely extraneous given the heterogeneity of the protocols employed, it is worth noting that BFR appears to induce minimal elevations of indirect markers of muscle damage while potentially showing signs of myocellular and vascular stress that attenuates over time. This finding is in line with other research [34] indicating BFR induces a repeated bout effect that is likely a product of myocellular upregulation of integral cellular defense mechanisms to attenuate the peri-exercise myocellular stress and post-exercise muscle damage release. However, the varied protocols included within this qualitative review limit strong conclusions regarding the acute- and longitudinal cellular changes that may occur following a HF-BFR protocol.

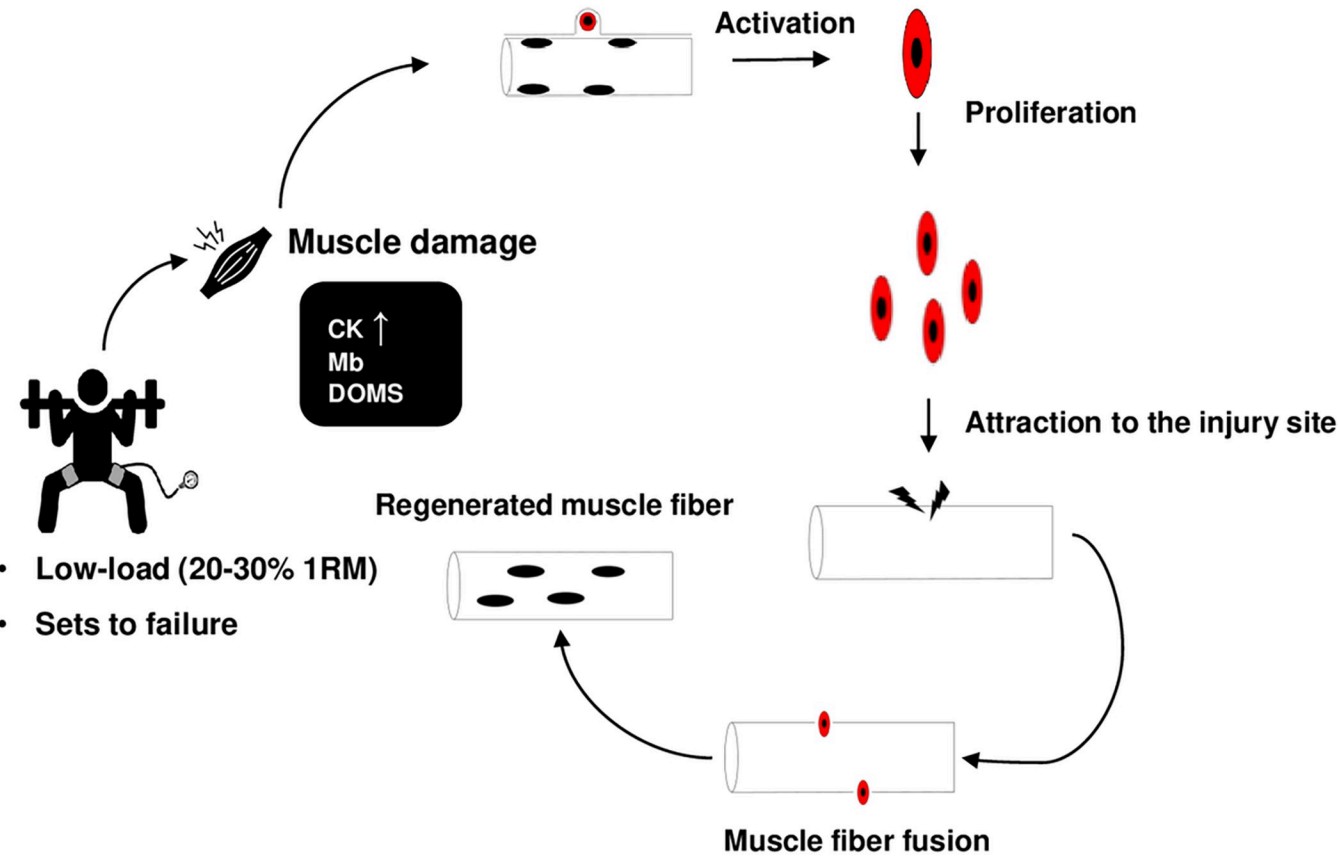

**Fig 4. Activation and proliferation of satellite cells after muscle damage induced by low–load resistance exercise with BFR.**

## 4.2 Limitations/Areas of concern identified in high-frequency BFR application

The present investigation has revealed significant limitations and areas of concern from the experimental high-frequency designs employed within the included studies of this scoping review. Highlighting limitations is important because the focus of a scoping review is to direct future research efforts by addressing said limitations [12].

Regardless of BFR exercise mode, personalized pressures were only used by one study in clinical populations [6]. Personalizing the pressure application has been recommended as standard of practice, particularly in rehabilitation settings [5]. Research has shown that there appears to be a minimal threshold of applied pressure (50% of arterial occlusion pressure, AOP) needed to accelerate fatigue accumulation during BFR training [35], at least in the lower extremity. As accelerated fatigue secondary to metabolite-induced effects on the myofiber has been proposed as a primary mechanism of BFR training [36], knowledge on the specific amounts of applied pressure is an important methodological consideration when integrating BFR training into practice. As only one study in our review incorporated personalized pressure applications, the relative intensity of the BFR stimulus during the exercise protocols is largely unknown despite the positive effects noted. Many reasons exist why personalized pressures may be ideal for research and clinical practice. Notably, personalized pressures allow for improved generalizability of the results of the study as it accounts for inter-individual differences in blood pressure, limb circumference and cuff widths used for BFR [5]. Studies have demonstrated that there appears to be differential perceptual and hemodynamic responses

when cuffs of different widths are inflated to the same arbitrary pressure [37], highlighting the importance of standardizing AOP. However, irrespective of cuff width, when standardized to AOP, perceptual [38] and hemodynamic responses at rest [39] appear to be equivocal indicating a similar restrictive stimulus. Therefore, extrapolation of the relative intensity of the BFR stimulus employed in the vast majority (95.4%) of the included studies is difficult and may have clinical/practical relevancy given the findings of Cerqueira et al. [35].

Only two studies included within our review investigated upper body exercise [25, 26]. Both were investigated during resistance training applications, leaving an absence of the potential effects of a high-frequency upper extremity aerobic exercise protocol. Within the two studies, qualitative heterogeneities exist, limiting generalizability to practice. One study [25] investigated the impact of 2 weeks of 12 sessions/week of non-failure (4 sets totaling 75 repetitions) BFR or no-BFR 30% 1RM bench press training on muscle size, strength, indirect markers of muscle damage and hormonal responses in 10 untrained men. The other study [26] had 13 men and women perform 6 sessions of biceps curls during 1 week of training at 30% 1RM in a similar non-failure protocol investigating muscle mass and strength. Both studies identified improvement in muscle strength and hypertrophy in the BFR condition, but not in the control conditions. None of the included studies investigated trained participants. This has relevancy given the proximal hypertrophy observed in the Yasuda et al. [25] study in the pectoralis major muscle. Recent research has hypothesized that volume of exercise performed [40] and training status [36] may play an important role in determining the magnitude of proximal hypertrophy observed during low-intensity BFR exercise. As both studies involved untrained participants in a non-failure protocol using two different exercise prescriptions, generalizability to the upper extremity in trained participants warrants caution.

The third major general limitation is the lack of women in the included studies. Women numbered only 8% (n = 30), limiting generalizability of the conclusions of the studies. Importantly, research has shown women exhibit greater sub-maximal endurance under blood flow restriction training compared to males [41], potentially under-dosing them to non-failure BFR training protocols when similar loads/intensities are used by both sexes. Five studies [7, 11, 17, 21, 26] in our review included both men and women but only two studies [17, 26] had a similar number of men as women. Sakamaki et al. [26] concluded in their 1-week biceps curl protocol that there were no sex-specific differences in training responses. However, women responded greater to the BFR during their luteal, but not follicular phase while there was a more uniform response from the men. Similarly, Iversen et al. [17] did not observe sex-specific differences in preservation of quadriceps mass following a 14-day high-frequency quadriceps strengthening protocol incorporating a fixed number of repetitions in post-surgical anterior cruciate ligament reconstruction patients. Thus, given the limited existing evidence, it is unclear whether females respond differently to men when a high-frequency BFR protocol is used despite a superior sub-maximal exercise capacity.

Only one study evaluated performance increases following high-frequency BFR resistance training application. The study in question [14] investigated the impact of 8 days of twice daily 3 sets of 15 repetitions of squats and leg curls performed at 20% 1RM in collegiate track and field athletes. The results indicated that BFR improved sprint but not jumping performance compared to work-matched, non-BFR exercise. The dearth of research investigating performance measures warrants careful consideration in future studies because as athletes continue to adopt BFR as part of their training regimens [42, 43], performance enhancement is a primary use case. As the literature on performance enhancement following high-frequency BFR training is limited to one study, future studies should integrate outcome measures to determine whether BFR can improve sports performance or accelerate recovery compared to both work-matched and effort-matched low- and high-intensity exercise.

The inclusion criteria for this scoping review stipulated that HF-BFR be performed for $> 4$ sessions per week. Within the 22 included studies, significant heterogeneities exist in terms of sessions per week, duration of the intervention and whether the exercise protocol was performed to failure or not. These prescription differences make comparing results between trials challenging. The duration of the interventions included in this review varied from 1 week (n = 5) to 3 weeks (n = 9) while sessions per week varied from 5 to 16 and almost half (n = 9) of the studies had participants in the BFR condition exercise to volitional fatigue. Importantly, the studies that have indicated stress to the myofiber and delayed supercompensation of myofiber CSA and muscle strength tended to have participants exercise to failure whereas the non-failure fixed repetition protocols responded beneficially without a delayed period of recovery. Future research is needed to help elucidate the optimal application parameters for producing beneficial musculoskeletal adaptations without a period of delayed supercompensation, as this delayed response likely reduces the potential feasibility of its practicality in athletes and other populations where sports performance is desired.

The included studies in this review are lacking in rigorous statistical analyses, mostly relying on null hypothesis (p-value) testing to determine significance between groups. Only five studies employed adjunctive statistical analyses in their investigations including 3 studies from the same author group [7, 10, 11] that included both confidence intervals and a power analysis, highlighting the potential for many of the included studies to be underpowered and/or overestimating the magnitude of the effect of the HF-BFR intervention. Employing other statistical approaches in conjunction with null hypothesis testing such as confidence intervals and effect sizes has been proposed to strengthen experimental results and improve confidence in the potential practical significance of the findings [44]. In particular, the absence of effect sizes prohibits understanding of the magnitude of the effect of HF-BFR while the largely absent confidence intervals increase the uncertainty surrounding the effect estimates. Therefore, despite the largely positive response of HF-BFR protocols, little can be gleaned from many of the investigations due to the absence of adjunctive statistical approaches. We recommend future studies attempt to include other statistical approaches that can better shape the potential effects of HF-BFR. These may include Bayesian statistics and modelling [45] or determination of minimally clinically important difference [46]. Both can help the BFR practitioner understand whether the integration of HF-BFR protocols is warranted in a short-term training program and if the benefits exceed potential risks and/or short-term performance decrements. In addition, prospective registration of clinical trials may aid in strengthening the current methodological shortcomings observed.

The last major area of concern we found is the absence of HF-BFR protocols on older adults and the very few studies (n = 4) [7, 14, 15, 17] on athletic and clinical populations. As these populations have been shown to derive significant benefits from the inclusion of BFR training [30, 47, 48], there is considerable uncertainty due to the absence of evidence on HF-BFR applications in these populations. When deciding to include HF-BFR protocols in cohorts with similar characteristics, caution is warranted.

## Specific resistance training major limitations/areas of concern

The current body of HF-BFR resistance training has two specific major limitations that we believe should be highlighted that may impact exercise prescription- a lack of adequate comparisons between low-load exercise on outcomes and the impact of HF-BFR failure training on recovery and adaptation profiles.

When looking to determine the specific effect of BFR resistance exercise, there are two common research designs that can provide information regarding the potential efficacy of

BFR. The first, adopted by a significant number of resistance training studies included within this review, used a work-matched design comparing low-intensity exercise with- and without BFR [14, 17, 22, 24, 27] or low-intensity BFR exercise compared to fixed repetition heavy load strength training [6, 7]. As the BFR prescriptions between protocols varied as well as the restrictive device and associated BFR stimulus due to the lack of a personalized pressure approach, proximity to failure is unknown [49]. Paired with the lack of adjunctive statistical approaches as mentioned in the above paragraph, it is difficult to determine the intensity of the exercise session and dissociate whether the effect of the intervention is due to the BFR stimulus, the volume performed, or both. This limitation is compounded when the HF-BFR protocols had participants exercise to failure during the BFR condition [8, 9, 16, 27, 28] and the control group performed a similar workload (e.g., matched repetitions) without restriction. As BFR accelerates fatigue accumulation at a given loading scheme, the BFR condition in those studies were exercising at the highest intensity whereas the work-matched low-load condition was likely nowhere near the same level of stress. As a result of the study designs employed comparing failure to non-failure work-matched protocols, a favorable adaptation profile was consistently observed in the HF-BFR condition compared to the low-intensity control condition (Table 2). Therefore, it is unknown whether the positive adaptations observed is a specific effect of BFR or the product of comparing failure- to non-failure exercise given longitudinal studies have shown similar musculoskeletal benefits when low-intensity exercise is performed to failure with- and without BFR in the upper and lower limbs [50, 51] as well as when comparing non-failure BFR to failure BFR protocols [52].

The second limitation we observed is that all the HF-BFR studies that resulted in delayed positive adaptations and/or elevated myocellular/vascular stress were performed to failure [7, 8, 11, 27]. This may have implications for integrating HF-BFR into practice as exercise to failure is inherently more stressful than work-matched non-BFR exercise. The current limited data suggests that when HF-BFR is performed to failure, there is a greater likelihood of a delayed supercompensation effect that could be observed greater than 10 days post-intervention [27]. Due to the limitations in the designs employed within this review, it is unknown whether similar results (e.g., delayed supercompensation) would have occurred if the low-load group without BFR exercised to failure. Nonetheless, it appears that if HF-BFR is integrated into practice, likely avoiding failure will reduce the potential for delayed supercompensation and improve its utility for those looking to maintain and/or increase relevant musculoskeletal outcomes of interest (e.g., muscle mass and strength) in shorter durations of time.

In addition, it must be considered that no study presented in this review compared HF-BFR training vs. low-frequency BFR training. Thus, it is not yet known whether lower frequency protocols (e.g., 2–3 times/week) could provide similar adaptations to high-frequency protocols in a similar time course. The only comparison presented in this regard was made in a meta-analysis [53], which identified that lower frequencies of BFR training would be better for eliciting muscle hypertrophy and strength increases.

## Specific aerobic training major limitations/areas of concern

Like HF-BFR resistance training, the current body of HF-BFR aerobic training has major limitations that may impact practical use of HF-BFR–homogeneity in exercise type (e.g., walking), lack of personalized prescriptions, and lack of continuous HF-BFR aerobic training applications.

Of the 22 included studies in this review, 5 employed aerobic exercise HF-BFR protocols and all used walking as the exercise type [4, 15, 18–20]. No study used other forms of aerobic

exercise that have been recommended for practice [5] such as cycling. Therefore, extrapolation of the study results to exercise types besides walking should be carefully considered.

All the aerobic studies analyzed did not attempt to personalize the intensity prescription in lieu of prescribing an arbitrary speed for exercise. Four studies [4, 18–20] had participants exercise using a set cadence of 50m/min whereas the other study used approximately 4km/h [15]. As BFR exercise has been proposed to be prescribed using intensities less than 50% VO$_2$max or heart rate reserve [5], it is unknown what relative intensity of exercise participants were performing when walking at arbitrary speeds. This limitation is like the above limitation with resistance training HF-BFR protocols because without standardizing the relative intensity of the efforts, it is challenging to determine what is a specific effect of the BFR stimulus. Recent research has shown that the internal load when performing aerobic exercise with 60% AOP BFR exceeds the same intensity of exercise performed without BFR despite reducing total work done during a 4-minute exercise bout [54]. Taken together, we cannot conclude what intensity of training is required to elicit the benefits observed in HF-BFR aerobic training protocols.

Last, there was a homogenous application of intermittent (e.g., where the cuffs are deflated during the rest periods and/or exercise is briefly stopped for a short duration of time) BFR pressure application across all our included studies. All studies adopted a similar framework– 5 sets of 2–3 minutes of interval walking separated by a 1-minute rest [4, 15, 18–20]. Another common BFR application that is used in practice is slightly longer, but continuous application of BFR where the exerciser is performing exercise for greater than 10 minutes per bout but typically not longer than 30 minutes [5]. As such, because no studies implemented this type of protocol, there is no conclusions that can be made regarding the potential efficacy of this type of prescription approach.

## 4.3 High-frequency BFR application future research suggestions

After reviewing the current literature base and highlighting some relevant limitations that preclude stronger extrapolations regarding the efficacy of HF-BFR protocols, we propose the following suggestions for future research (Table 3). We feel that future studies should focus on these questions to help fill in the gaps in understanding and provide important insights into the potential benefits of incorporating a HF-BFR protocol into a training or rehabilitation program.

**Table 3. Current limitations in high–frequency blood flow restriction training literature and suggestions for future studies.**

| Observed Current Limitation | Relevant Research Question | Future Research Suggestions to Address Current Limitations |
|---|---|---|
| Lack of personalized pressure applications | What is the magnitude of observed effects in HF-BFR protocols when pressures are personalized (e.g., % AOP)? | Utilize a personalized pressure application for BFR prescription instead of arbitrary pressures |
| Lack of upper extremity investigations | Do upper body exercises respond similarly to a HF-BFR protocol as lower body exercise? | Use upper body exercises |
| Lack of females | Is there a differential response in females undergoing a HF-BFR protocol than males? | Include more females |
| No studies in the older adults/elderly populations | Do older adults respond similarly as other populations to a HF-BFR protocol? | Include elderly adults |
| Limited studies on athletes | Is there a positive benefit (on muscle hypertrophy, strength and/or recovery/performance) for incorporating HF-BFR in athletic populations? | Include athletic populations |

(*Continued*)

**Table 3.** (Continued)

| Observed Current Limitation | Relevant Research Question | Future Research Suggestions to Address Current Limitations |
|---|---|---|
| Limited studies on clinical populations | Does the addition of HF-BFR to a rehabilitation program accelerate post-surgical and/or post-injury recovery over traditional rehabilitation? | Include clinical populations |
| Absence of studies comparing different HF-BFR protocols | Is there a meaningful difference between the number of HF-BFR sessions per week on relevant musculoskeletal and/or performance outcomes? | Compare 5 sessions per week to 10–12 sessions per week |
| Limited adjunctive statistical approaches used/ potentially underpowered study designs | Is the addition of a HF-BFR training program provide practically meaningful changes in relevant musculoskeletal and/or functional outcomes? | Incorporate a priori power analyses to ensure adequate sample sizes as well as adjunctive statistical approaches (e.g., confidence intervals, effect sizes, MCID) |
| Lack of adequate comparison to traditionally recommended weekly BFR frequencies | Does HF-BFR training outperform traditionally recommended weekly frequencies (2-3x/week) in relevant musculoskeletal outcome measure? | Utilize a study design comparing 3 weeks of HF-BFR (> 4 sessions/week) to 3 weeks of 2-3x/week |
| Uncertainty regarding the benefit of HF-BFR compared to low-load effort-matched exercise | When both low-load and low-load BFR conditions are performed to failure, does the BFR condition outperform low-load exercise alone in relevant musculoskeletal outcomes? | Utilize a study design where low-load exercise and low-load exercise with BFR are compared with both groups exercising to failure during a high-frequency training program; Investigate myocellular responses (e.g., satellite cells) to elucidate whether BFR induces a specific effect or not compared to low-load training at the same intensity |
| Limited data regarding the proximal effects of HF-BFR | Is there a benefit for muscles located proximally to the restrictive stimulus when performing a HF-BFR program? | Incorporate measurement of muscle mass and strength for muscle groups located proximal to the restrictive site |
| Absence of data investigating hypoalgesia response to HF-BFR | Does the addition of a HF-BFR training program enhance post-exercise hypoalgesia compared to the same exercise prescription without BFR? | Incorporate algometry and other associated measures used to assess post-exercise hypoalgesia |
| Limited data regarding the impact of failure exercise on post-exercise recovery | Does the addition of a HF-BFR training program to failure increase the likelihood of experiencing delayed supercompensation? | Compare failure to non-failure (fixed repetition) HF-BFR protocols on relevant markers of recovery and/or delayed supercompensation |
| Limited data regarding the vascular response to HF-BFR | What participant and/or BFR exercise prescription factors impact the likelihood of experiencing vascular adaptations during a HF-BFR protocol? | Compare different participant and/or BFR prescription types on vascular adaptations during HF-BFR |
| Absence of data regarding post-intervention hypotensive response to HF-BFR | Does the addition of HF-BFR enhance the post-exercise hypotensive response acutely and impact resting blood pressure values? | Include monitoring of blood pressure responses during HF-BFR protocols at rest and following exercise |
| *RT only*: Absence of data regarding the impact of intermittent BFR (e.g., where the cuff is inflated during the exercise but deflated during the rest period) pressure application on HF-BFR protocols | Does intermittent BFR confer similar benefits to continuous application of BFR during a HF-BFR protocol? | Compare the same workload with- and without continuous BFR pressure application during a HF-BFR protocol |
| *AT only*: Absence of relative intensity exercise prescriptions (e.g., %HRR or VO2max) | Is there a specific intensity required to elicit positive aerobic training adaptations during a HF-BFR protocol? | Investigate HF-BFR aerobic training protocols using a %HRR or %VO2max, avoiding using arbitrary speed prescriptions |
| *AT only*: Absence of continuous aerobic training studies | Does continuous aerobic exercise produce superior benefits on relevant markers of aerobic training adaptations to the same exercise performed without BFR? | Include protocols that have participants exercise for a set period continuously (e.g., 15 minutes) in a high-frequency program with- and without BFR. |

AOP: Arterial occlusion pressure; AT: aerobic training; BFR: blood flow restriction; HF–BFR: high–frequency BFR (> 4 sessions/week); HRR: heart rate reserve; MCID: minimum clinically important difference; RT: resistance training.

# 5 Conclusions

Despite significant heterogeneities in the current body of evidence, it appears that HF-BFR is a potentially viable solution to improving muscle strength and muscle mass despite some studies indicating myocellular and vascular stress and/or a delayed supercompensation effect. Due to

the varied study designs, extrapolation of general efficacy is challenging; however, despite these limitations, HF-BFR does appear to be a strategy to optimize adaptations in a shorter time. More research is needed to clarify the magnitude of benefits that HF-BFR may provide compared to low- and heavy load strength training and the time course of the adaptation profiles. Nonetheless, despite one study indicating an adverse event in a failure protocol that resolved without medical intervention, HF-BFR appears to be safe and well tolerated in a variety of prescriptions in healthy, athletic and/or clinical populations.

## Supporting information

**S1 Checklist. PRISMA-ScR.**
(DOCX)

**S1 Data.**
(XLSX)

## Author Contributions

**Conceptualization:** Victor Sabino de Queiros, Nicholas Rolnick, Breno Guilherme de Araújo Tinôco Cabral.

**Data curation:** Victor Sabino de Queiros, Phelipe Wilde de Alcântara Varela, Paulo Moreira Silva Dantas.

**Formal analysis:** Victor Sabino de Queiros.

**Methodology:** Victor Sabino de Queiros, Nicholas Rolnick, Phelipe Wilde de Alcântara Varela, Paulo Moreira Silva Dantas.

**Project administration:** Paulo Moreira Silva Dantas.

**Supervision:** Breno Guilherme de Araújo Tinôco Cabral, Paulo Moreira Silva Dantas.

**Writing – original draft:** Victor Sabino de Queiros, Nicholas Rolnick.

**Writing – review & editing:** Victor Sabino de Queiros, Nicholas Rolnick, Phelipe Wilde de Alcântara Varela, Breno Guilherme de Araújo Tinôco Cabral, Paulo Moreira Silva Dantas.

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
