## [Decision Letter · Decision Letter 0]

19 Oct 2022

PONE-D-22-22090Physiological adaptations and myocellular stress in short-term, high-frequency blood flow restriction training: A scoping reviewPLOS ONE

Dear Dr. de Queiros,

Thank you for submitting your manuscript to PLOS ONE. After careful consideration, we feel that it has merit but does not fully meet PLOS ONE’s publication criteria as it currently stands. Therefore, we invite you to submit a revised version of the manuscript that addresses the points raised during the review process.

We look forward to receiving your revised manuscript.

Kind regards,

Jeremy P Loenneke

Academic Editor

PLOS ONE

Journal Requirements:

2. Please complete a PRISMA-ScR checklist (available at https://www.equator-network.org/wp-content/uploads/2018/09/PRISMA-ScR-Fillable-Checklist-1.docx) and upload it as supplementary file.

   "VSQ and PWAV were financed in part by a scholarship from the Coordenação de Aperfeiçoamento de Pessoal de Nível Superior (CAPES), Brazil - finance code 001."

   "NR is the founder of THE BFR PROS, a BFR education company that provides BFR training workshops to fitness and rehabilitation professionals across the world using a variety of BFR devices. NR has no financial relationships with any cuff manufacturers/distributors. The other authors declare no potential or actual conflicts of interest."

6. We note that this manuscript is a systematic review or meta-analysis; our author guidelines therefore require that you use PRISMA guidance to help improve reporting quality of this type of study. Please upload copies of the completed PRISMA checklist as Supporting Information with a file name “PRISMA checklist”.

Additional Editor Comments:

Both reviewers found merit but suggestions for improvement have been noted. Please consider each reviewers comments when revising. Pay particular attention to Reviewer 2 who suggested that the manuscript could be more focused in areas.

Reviewers' comments:

Reviewer's Responses to Questions

**Comments to the Author**

1. Is the manuscript technically sound, and do the data support the conclusions?

Reviewer #1: Yes

Reviewer #2: Partly

2. Has the statistical analysis been performed appropriately and rigorously? 

Reviewer #1: N/A

Reviewer #2: Yes

3. Have the authors made all data underlying the findings in their manuscript fully available?

Reviewer #1: Yes

Reviewer #2: Yes

4. Is the manuscript presented in an intelligible fashion and written in standard English?

Reviewer #1: Yes

Reviewer #2: Yes

5. Review Comments to the Author

Reviewer #1: de Queiros and colleagues provide an in-depth overview of myocellular adaptations to high frequency blood flow restricted exercise training. Authors focus on myocellular stress, and they provide a great overview of recent literature in this field. This overarching review is well written and referenced. I support publication of the review and have only a minor comment.

Given the emphasis on myocellular stress, a figure collating some of the data on satellite cells, muscle damage, inflammation would add novelty to the manuscript. The provided figures largely reflect muscle size and strength and have been shown many times before. If a summary figure is difficult to tabulate given the limited studies directly assessing myocellular stress, perhaps a graphical abstract of sorts could be created linking some of the ideas proposed by reviewers. Acute inflammation can activate satellite cells, promoting myonuclear accrual, etc. and how BFR specifically can initiate these signaling cascades. This would be a great reference tool for researchers and clinicians seeking to understand or explain potential risks of high frequency BFR.

Reviewer #2: Authors conducted an interesting scoping review examining adaptations to high frequency BFR training over short durations. I can tell a lot of time and work went into this manuscript. I do wonder if this manuscript may benefit from a more narrow focus? For example, the limitations at the end were longer that the discussion of findings. Overall, the findings appear quite heterogenous in nature. This is not surprising as methods employed and control groups utilized are different across the majority of studies included in the present review. I thank the authors for the opportunity to review your work and I have provided feedback below which I hope you find helpful.

Introduction:

Lines 53-54: “In addition, low-intensity aerobic training programs with BFR can promote muscle hypertrophy, increases in lower limb strength, and aerobic capacity despite low intensities of training [3,4].” - - - Is this relative to a non-BFR aerobic control?

Intro, paragraph 2: In general, this paragraph seems to lack focus. Some ideas on frequency are presented…but I believe it can be developed some. For example, authors reference powerlifters at the end, but what point is being made with this inclusion? “…appears to maximize hypertrophic adaptions in well trained powerlifting athletes” demonstrating that the addition of BFR may provide…..

Intro, paragraph 2: “High-frequency BFR training can be a potentially useful tool to accelerate recovery in clinical rehabilitation settings, since the technique can provide positive physiological adaptations in less time compared to high-load strength training [6].” - - If it is high frequency is it really less time? Or is this compared to a high load training program of similar training frequency?

In it’s current form the introduction does not provide a compelling rationale for the present manuscript. For example authors primarily provide evidence from Landow and Nielson as examples of discrepant findings in strength outcomes between high frequency BFR programs. However, I would argue that this is true in the majority of the BFR literature regardless of frequency. Increases in 1RM strength are generally inconsistent across the BFR literature. However, when performed with low loads, BFR will typically provide similar changes in muscle size. It would be more compelling to provide muscle growth data (if available).

Discussion:

Lines 520-522: “Regardless of exercise mode, sessions per week (> 4), and method of assessment (e.g., maximal isometric voluntary contraction), it appears that HF-BFR training generally produces superior increases in muscle strength when compared to the same intensity exercise performed without BFR.” – What do authors mean by “same intensity” Some would suggest this would refer to the proximity to failure. I believe that the authors are referring to same “relative training load (%1RM or same RM). However, it would be important to specify both proximity to failure and %load/RM.

Lines 530-532: “Intriguingly, the results appear to indicate that HF-BFR application may produce similar gains in muscle strength as longer duration protocols (> 3 weeks) in less time despite differences in exercise mode, population studied and BFR application parameters. - - Given the large range in strength changes reported, I am not sure this is as clear as the authors suggest. Again, I would encourage authors to consider the current consensus on strength adaptations to low load alternatives of training. They are typically difficult to predict and typically underperform compared to high load alternatives. Authors indicate that strength outperformed the comparison group 78% of the time….however what that “comparison group” was is really important for the interpretation of what this means.

Not much discussion of provided on muscle size other than a comparison to longer duration values. However, this is (as the authors mention) not the purpose of the present manuscript. It may be more important to discuss why some had larger and others had smaller changes in muscle size. It may also be worthwhile to discuss differences in control conditions across studies. For example, it appears that many of the control groups were volume matched. This may simply indicate the control groups were underdosed…as low loads performed to failure typically produce similar growth as low loads with BFR to failure (although BFR conditions will most typically perform less volume). If there was growth what does this indicate? What are the implications of this finding?

I think one issue with the present interpretation is that no study (unless I missed it) compared low/regular frequency BFR over the same period of time (i.e., 3-4 weeks) to the high frequency BFR condition. Thus, it is unclear if there really is a benefit of the increased frequency compared to more common frequencies of training. This is found later in the limitations section…but it does make it difficult to recommend high frequency BFR training when it is unknown if it is better than a lower frequency option.

After re-reading this manuscript it seems to outline (through the limitations sections) barriers and issues with interpreting the high frequency BFR literature when considering its incorporation into clinical practice. However, the introduction provides a different expectation. I would work on overall flow to increase the cohesiveness of this manuscript. I would recommend expanding the discussion on the findings of this investigation while reducing focus on limitations. I may be wrong on this recommendation and encourage the authors to consider all feedback along with your own thoughts/opinion.

6. PLOS authors have the option to publish the peer review history of their article (what does this mean?). If published, this will include your full peer review and any attached files.

Reviewer #1: No

Reviewer #2: No

---

## [Author Response · Author response to Decision Letter 0]

24 Oct 2022

REVIEWER 1

Given the emphasis on myocellular stress, a figure collating some of the data on satellite cells, muscle damage, inflammation would add novelty to the manuscript. The provided figures largely reflect muscle size and strength and have been shown many times before. If a summary figure is difficult to tabulate given the limited studies directly assessing myocellular stress, perhaps a graphical abstract of sorts could be created linking some of the ideas proposed by reviewers. Acute inflammation can activate satellite cells, promoting myonuclear accrual, etc. and how BFR specifically can initiate these signaling cascades. This would be a great reference tool for researchers and clinicians seeking to understand or explain potential risks of high frequency BFR.

Response: We added a figure to our discussion. Introducing the relationship between muscle damage and satellite cell activation. We hope we have responded to the reviewer's request.

REVIEWER 2

Introduction:

Lines 53-54: “In addition, low-intensity aerobic training programs with BFR can promote muscle hypertrophy, increases in lower limb strength, and aerobic capacity despite low intensities of training [3,4].” - - - Is this relative to a non-BFR aerobic control?

Response: Perfect! Good observation. Yes, this is relative to a non-BFR aerobic control. We emphasize this aspect in this new version (Lines 56-57). Thank you very much!

Intro, paragraph 2: In general, this paragraph seems to lack focus. Some ideas on frequency are presented…but I believe it can be developed some. For example, authors reference powerlifters at the end, but what point is being made with this inclusion? “…appears to maximize hypertrophic adaptions in well trained powerlifting athletes” demonstrating that the addition of BFR may provide…

Response: Perfect! Good observation. In this paragraph, we seek to present high-frequency and short-term blood flow restriction training and its possible applicability. We agree that the paragraph could be improved. As requested by the reviewer, we have added additional information regarding the applicability of high-frequency blood flow restriction training for highly trained athletes (Lines 67-69). We hope we have complied with the reviewer's request. Thank you very much!

Intro, paragraph 2: “High-frequency BFR training can be a potentially useful tool to accelerate recovery in clinical rehabilitation settings, since the technique can provide positive physiological adaptations in less time compared to high-load strength training [6].” - - If it is high frequency is it really less time? Or is this compared to a high load training program of similar training frequency?

Response: The statement concerns the capacity of high-frequency training with blood flow restriction to promote short-term adaptations.

In it’s current form the introduction does not provide a compelling rationale for the present manuscript. For example authors primarily provide evidence from Landow and Nielson as examples of discrepant findings in strength outcomes between high frequency BFR programs. However, I would argue that this is true in the majority of the BFR literature regardless of frequency. Increases in 1RM strength are generally inconsistent across the BFR literature. However, when performed with low loads, BFR will typically provide similar changes in muscle size. It would be more compelling to provide muscle growth data (if available).

Response: Perfect! We added information related to the differences presented about the muscle fiber size (Line 82-84). However, we believe that it is important to provide information related to strength tests, especially because we present evidence of strength decline after high-frequency blood flow restriction training. 

Discussion:

Lines 520-522: “Regardless of exercise mode, sessions per week (> 4), and method of assessment (e.g., maximal isometric voluntary contraction), it appears that HF-BFR training generally produces superior increases in muscle strength when compared to the same intensity exercise performed without BFR.” – What do authors mean by “same intensity” Some would suggest this would refer to the proximity to failure. I believe that the authors are referring to same “relative training load (%1RM or same RM). However, it would be important to specify both proximity to failure and %load/RM.

Response: Great! In this new version, we emphasize that the intensity concerns the external load (Line 533). Thank you very much!

Lines 530-532: “Intriguingly, the results appear to indicate that HF-BFR application may produce similar gains in muscle strength as longer duration protocols (> 3 weeks) in less time despite differences in exercise mode, population studied and BFR application parameters. - - Given the large range in strength changes reported, I am not sure this is as clear as the authors suggest. Again, I would encourage authors to consider the current consensus on strength adaptations to low load alternatives of training. They are typically difficult to predict and typically underperform compared to high load alternatives. Authors indicate that strength outperformed the comparison group 78% of the time….however what that “comparison group” was is really important for the interpretation of what this means.

Response: Perfect! In this new version, we specify that the information presented refers to comparisons between BFR low-intensity versus non-BFR low-intensity. In addition, we present results regarding BFR low-load versus non-BFR high-load comparisons. 

Not much discussion of provided on muscle size other than a comparison to longer duration values. However, this is (as the authors mention) not the purpose of the present manuscript. It may be more important to discuss why some had larger and others had smaller changes in muscle size. It may also be worthwhile to discuss differences in control conditions across studies. For example, it appears that many of the control groups were volume matched. This may simply indicate the control groups were underdosed…as low loads performed to failure typically produce similar growth as low loads with BFR to failure (although BFR conditions will most typically perform less volume). If there was growth what does this indicate? What are the implications of this finding?

I think one issue with the present interpretation is that no study (unless I missed it) compared low/regular frequency BFR over the same period of time (i.e., 3-4 weeks) to the high frequency BFR condition. Thus, it is unclear if there really is a benefit of the increased frequency compared to more common frequencies of training. This is found later in the limitations section…but it does make it difficult to recommend high frequency BFR training when it is unknown if it is better than a lower frequency option.

After re-reading this manuscript it seems to outline (through the limitations sections) barriers and issues with interpreting the high frequency BFR literature when considering its incorporation into clinical practice. However, the introduction provides a different expectation. I would work on overall flow to increase the cohesiveness of this manuscript. I would recommend expanding the discussion on the findings of this investigation while reducing focus on limitations. I may be wrong on this recommendation and encourage the authors to consider all feedback along with your own thoughts/opinion.

Response: We understand the reviewer's concern, however, we should point out that we are dealing with a scoping review. In accordance with the recommendations for scoping review, the purpose of the discussion is to present the main results and discuss limitations and guide future research efforts. In our conception, we believe we have complied with this proposal. We would like to point out that our focus was to provide an overview of training with high-frequency blood flow restriction, including the limitations of the studies that analyzed this training methodology so that future research could be better equipped to conduct a rigorous trial and avoid the pitfalls of the past prescriptions. We have added a number of references to the importance of discussing limitations, as well as a supportive reference (12).

---

## [Decision Letter · Decision Letter 1]

15 Nov 2022

PONE-D-22-22090R1Physiological adaptations and myocellular stress in short-term, high-frequency blood flow restriction training: A scoping reviewPLOS ONE

Dear Dr. de Queiros,

Thank you for submitting your manuscript to PLOS ONE. After careful consideration, we feel that it has merit but does not fully meet PLOS ONE’s publication criteria as it currently stands. Therefore, we invite you to submit a revised version of the manuscript that addresses the points raised during the review process.

Reviewers found your manuscript acceptable with a few additional suggested changes. I think the suggested changes will help clarify details for future readers.

We look forward to receiving your revised manuscript.

Kind regards,

Jeremy P Loenneke

Academic Editor

PLOS ONE

Journal Requirements:

Reviewers' comments:

Reviewer's Responses to Questions

**Comments to the Author**

1. If the authors have adequately addressed your comments raised in a previous round of review and you feel that this manuscript is now acceptable for publication, you may indicate that here to bypass the “Comments to the Author” section, enter your conflict of interest statement in the “Confidential to Editor” section, and submit your "Accept" recommendation.

Reviewer #1: All comments have been addressed

Reviewer #2: All comments have been addressed

2. Is the manuscript technically sound, and do the data support the conclusions?

Reviewer #1: Yes

Reviewer #2: Yes

3. Has the statistical analysis been performed appropriately and rigorously? 

Reviewer #1: N/A

Reviewer #2: Yes

4. Have the authors made all data underlying the findings in their manuscript fully available?

Reviewer #1: Yes

Reviewer #2: (No Response)

5. Is the manuscript presented in an intelligible fashion and written in standard English?

Reviewer #1: Yes

Reviewer #2: Yes

6. Review Comments to the Author

Reviewer #1: The authors have addressed all my comments; I appreciate the new review figure. Congratulations on a fantastic review.

Reviewer #2: Great work on this manuscript. Clearly a lot of work has gone into this paper. I belie that this version is greatly improved. I have some additional comments which I hope may strengthen the manuscript.

1. Introduction:

I found the introduction to be much improved on the latest revision. Thank you for the changes that have been made.

2. Results:

MVC section line 243: Are authors referencing exclusively low load training controls without BFR? “All studies included analyses of the isometric strength of the knee extensors and only one did not identify a significant increase after BFR training [24], while no study identified increases in isometric strength after non-BFR training.” - -∫I would add an additional descriptor here.

3. Throughout the results authors are sometimes specific what the control group is and other times will refer to as a control.

4. Muscle endurance results section: Were “differences between groups washed out after 12 weeks despite continued training? After 4 weeks post-surgery, the high-frequency BFR training group exhibited ~50% greater sub-maximal quadriceps endurance than the non-BFR work matched control group, but the differences between groups washed out after 12 weeks.

5. Jump and sprint section: BFR produced superior decreases relative to what?

“The results of this study indicate BFR produced superior decreases in 30-m sprint time that was attributed to improvements in early acceleration during the first 10-m.”

6. Muscle size lines 307-309: Was the control a low load intervention? “From the 2nd to the 14th post-surgery day, a group of patients underwent an BFR exercise intervention, while a control group underwent an exercise intervention without BFR.”

7. Microscopic measurements section lines 351-353: …in the control condition which performed…

blocks of high-frequency BFR training versus heavy load training without BFR in a traditional high

load training routine; in the BFR condition, there was a significant increase in CSA of type I fibers, but

not of type II fibers; in the control condition, no significant differences were reported.

8. Inflammatory responses lines 473-474: Was this low load? I would add low load after high frequency. “One study [28] analyzed macrophage content by immunofluorescence before, after eight days, and three and ten days after three weeks of high-frequency training with and without BFR.”

9. Vascular adaptations lines 496-497: Was this low load? One study [8] analyzed the number of capillaries per muscle fiber and changes in the perivascular basement membrane in high-frequency training with and without BFR.

10. It may seem redundant to the authors but it may be most appropriate to add “low load” whenever discussing “resistance training with BFR” as this provides further detail which may be important to the reader.

11. Discussion lines 570-573: How was edema measured? “Their results seemed to indicate that the

hypertrophic response to BFR occurred without the presence of edema (in 3 weeks) whereas in high

load training, concomitant edema occurs (at week 3), impairing the initial adaptation process yet over

the same number of sessions, tended to induce greater muscle hypertrophy and muscle strength (at 6 weeks).”

12. Limitations section lines 623-625: Many studies have actually employed 40% AOP…so 40 may be more appropriate than 50% - - “Research has shown that there appears to be a minimal threshold of applied pressure (50% of arterial occlusion pressure, AOP) needed to accelerate fatigue accumulation during BFR training [35],..”

7. PLOS authors have the option to publish the peer review history of their article (what does this mean?). If published, this will include your full peer review and any attached files.

Reviewer #1: **Yes: **Christopher Fry

Reviewer #2: No

---

## [Author Response · Author response to Decision Letter 1]

17 Nov 2022

REVIEWER 2

1. Introduction:

I found the introduction to be much improved on the latest revision. Thank you for the changes that have been made.

Response: Thanks for the help provided!

2. Results:

MVC section line 243: Are authors referencing exclusively low load training controls without BFR? “All studies included analyses of the isometric strength of the knee extensors and only one did not identify a significant increase after BFR training [24], while no study identified increases in isometric strength after non-BFR training.” - -∫I would add an additional descriptor here.

Response: Perfect! We emphasize that we are referring to low load/intensity training. Thank you very much! (Line: 246)

3. Throughout the results authors are sometimes specific what the control group is and other times will refer to as a control.

Response: In this new version, we adopted only "comparator". Thanks for the comment.

4. Muscle endurance results section: Were “differences between groups washed out after 12 weeks despite continued training? After 4 weeks post-surgery, the high-frequency BFR training group exhibited ~50% greater sub-maximal quadriceps endurance than the non-BFR work matched control group, but the differences between groups washed out after 12 weeks.

Response: Thanks for the comment. As specified, BFR training was applied before surgery. After surgery, participants were involved in a rehabilitation program (without BFR training). We point out this aspect in this new version. (Line: 279-280)

5. Jump and sprint section: BFR produced superior decreases relative to what?

“The results of this study indicate BFR produced superior decreases in 30-m sprint time that was attributed to improvements in early acceleration during the first 10-m.”

Response: Good observation! We refer to low-load training without BFR. We highlight this point in this new version. Thank you very much! (Line: 290-291).

6. Muscle size lines 307-309: Was the control a low load intervention? “From the 2nd to the 14th post-surgery day, a group of patients underwent an BFR exercise intervention, while a control group underwent an exercise intervention without BFR.”

Response: Good observation! The study in question analyzed exercise without external overload. We highlight this aspect in this new version. Thank you very much! (Line: 312-313).

7. Microscopic measurements section lines 351-353: …in the control condition which performed…

blocks of high-frequency BFR training versus heavy load training without BFR in a traditional high load training routine; in the BFR condition, there was a significant increase in CSA of type I fibers, butnot of type II fibers; in the control condition, no significant differences were reported.

Response: Perfect! We made the change suggested by the reviewer! Thank you very much! (Line: 360)

8. Inflammatory responses lines 473-474: Was this low load? I would add low load after high frequency. “One study [28] analyzed macrophage content by immunofluorescence before, after eight days, and three and ten days after three weeks of high-frequency training with and without BFR.”

Response: Good observation! We point out that the intervention tested low-load training. (Line: 485).

9. Vascular adaptations lines 496-497: Was this low load? One study [8] analyzed the number of capillaries per muscle fiber and changes in the perivascular basement membrane in high-frequency training with and without BFR.

Response: Good observation! We point out that the intervention tested low-load training. (Line: 509).

10. It may seem redundant to the authors but it may be most appropriate to add “low load” whenever discussing “resistance training with BFR” as this provides further detail which may be important to the reader.

Response: Perfect! When necessary, we add “low-load” to discussions around resistance training with BFR. Changes are highlighted throughout the text.

11. Discussion lines 570-573: How was edema measured? “Their results seemed to indicate that the hypertrophic response to BFR occurred without the presence of edema (in 3 weeks) whereas in high load training, concomitant edema occurs (at week 3), impairing the initial adaptation process yet over the same number of sessions, tended to induce greater muscle hypertrophy and muscle strength (at 6 weeks).”

Response: Good observation. We present the method used in the study to identify edema. Thank you very much! (Line: 580-581)

12. Limitations section lines 623-625: Many studies have actually employed 40% AOP…so 40 may be more appropriate than 50% - - “Research has shown that there appears to be a minimal threshold of applied pressure (50% of arterial occlusion pressure, AOP) needed to accelerate fatigue accumulation during BFR training [35],..”

Response: Good observation. However, the information presented is supported by the cited article. In this sense, we chose to keep it at 50%.

---

## [Decision Letter · Decision Letter 2]

15 Dec 2022

Physiological adaptations and myocellular stress in short-term, high-frequency blood flow restriction training: A scoping review

PONE-D-22-22090R2

Dear Dr. de Queiros,

We’re pleased to inform you that your manuscript has been judged scientifically suitable for publication and will be formally accepted for publication once it meets all outstanding technical requirements.

Kind regards,

Jeremy P Loenneke

Academic Editor

PLOS ONE

Additional Editor Comments (optional):

Reviewers' comments:

Reviewer's Responses to Questions

**Comments to the Author**

1. If the authors have adequately addressed your comments raised in a previous round of review and you feel that this manuscript is now acceptable for publication, you may indicate that here to bypass the “Comments to the Author” section, enter your conflict of interest statement in the “Confidential to Editor” section, and submit your "Accept" recommendation.

Reviewer #2: All comments have been addressed

2. Is the manuscript technically sound, and do the data support the conclusions?

Reviewer #2: Yes

3. Has the statistical analysis been performed appropriately and rigorously? 

Reviewer #2: Yes

4. Have the authors made all data underlying the findings in their manuscript fully available?

Reviewer #2: (No Response)

5. Is the manuscript presented in an intelligible fashion and written in standard English?

Reviewer #2: Yes

6. Review Comments to the Author

Reviewer #2: Authors have addressed all of my major concerns. I have have enjoyed providing feedback on this manuscript.

7. PLOS authors have the option to publish the peer review history of their article (what does this mean?). If published, this will include your full peer review and any attached files.

Reviewer #2: No

---

## [Editor Report · Acceptance letter]

20 Dec 2022

PONE-D-22-22090R2 

Physiological adaptations and myocellular stress in short-term, high-frequency blood flow restriction training: A scoping review 

Dear Dr. de Queiros:

I'm pleased to inform you that your manuscript has been deemed suitable for publication in PLOS ONE. Congratulations! Your manuscript is now with our production department. 

Kind regards, 

on behalf of

Dr. Jeremy P Loenneke 

Academic Editor

PLOS ONE